# Parental experiences orchestrate locust egg hatching synchrony by regulating nuclear export of precursor miRNA

Ya'nan Zhu [1,2,4], Jing He[1,4], Jiawen Wang[1], Wei Guo [1], Hongran Liu[1], Zhuoran Song[1] & Le Kang [1,2,3] ✉

Parental experiences can affect the phenotypic plasticity of offspring. In locusts, the population density that adults experience regulates the number and hatching synchrony of their eggs, contributing to locust outbreaks. However, the pathway of signal transmission from parents to offspring remains unclear. Here, we find that transcription factor Forkhead box protein N1 (FOXN1) responds to high population density and activates the *poly-pyrimidine tract-binding protein 1* (*Ptbp1*) in locusts. FOXN1–PTBP1 serves as an upstream regulator of miR-276, a miRNA to control egg-hatching synchrony. PTBP1 boosts the nucleo-cytoplasmic transport of pre-miR-276 in a "CU motif"-dependent manner, by collaborating with the primary exportin protein exportin 5 (XPO5). Enhanced nuclear export of pre-miR-276 elevates miR-276 expression in terminal oocytes, where FOXN1 activates *Ptbp1* and leads to egg-hatching synchrony in response to high population density. Additionally, PTBP1-prompted nuclear export of pre-miR-276 is conserved in insects, implying a ubiquitous mechanism to mediate transgenerational effects.

The phenomenon that parental life experiences can influence phenotypes in their offspring has tremendous implications for basic biology. Accumulating evidence from worms to humans shows that the environment experienced by parents can influence phenotypes across generations[1]. Furthermore, parental effects are most pronounced in the early postnatal period[2]. Population density, as an environmental factor, also can trigger transgenerational effects. Female animals in response to high-density conditions tend to alter the traits of offspring including body size[3], growth rate[4], and aggressive behavior[5]. The migratory locust (*Locusta migratoria*), a worldwide pest, displays density-dependent phenotypic plasticity[6] in the present and the next generation. Gregarious and solitarious locusts can reciprocally transform between each other in response to population density changes. High-density gregarious and low-density solitarious locusts show significant differences in behavior, body color, immunity, hormones and pheromones, and flight ability regulated by both proteins and non-coding RNAs[7–13]. In particular, the experiences of different population densities result in divergence of the reproductive strategies in gregarious and solitarious locusts. Solitarious locusts lay more and smaller-sized eggs with more irregular hatching, while gregarious locusts lay fewer and larger-sized eggs that hatch more synchronously[14–18], aiding in swarm formation in the next generation. Focused evaluation of such transgenerational effects may reveal a non-canonical mechanism of inheritance.

The adaptive transgenerational effects resulting from environmental experiences are complicated and involve various cytoplasmic factors in the egg, such as mitochondria, yolk amount, hormones, and mRNAs[19–22], to shape offspring phenotype and fitness across multiple generations. In the migratory locust, density-related transgenerational effects are mediated by a number of gene expression and signal pathways. For instance, differential expression of *Syntaxin 1A* in solitarious and gregarious females affects the size variation of progeny

[1]State Key Laboratory of Integrated Management of Pest Insects and Rodents, Institute of Zoology, Chinese Academy of Sciences, Beijing 100101, China. [2]University of Chinese Academy of Sciences, Beijing 100101, China. [3]College of Life Science, Hebei University, Baoding, Hebei 071002, China. [4]These authors contributed equally: Ya'nan Zhu, Jing He. ✉e-mail: lkang@ioz.ac.cn

eggs by regulating the yolk proteins in oocytes and eggs[15]. Furthermore, the differential expression of *heat-shock proteins* (*Hsps*) in parents can transmit to progeny eggs, and is closely related to egg size variations between gregarious and solitarious locusts[14]. In addition, the different cold hardiness of eggs from gregarious and solitarious locusts may be regulated by the stimulated lipid metabolism, heat-shock proteins, and carboxylic acid transport pathways[23]. Thus, these significant advances pave the way for further intensive studies of the mechanisms underlying density-regulated parental effects in locusts.

Recent studies have confirmed small non-coding RNAs as potential mediators of the transmission of environmental information through germlines[24,25]. Notably, the mechanism by which non-genetic factors such as small RNAs are induced by the environment represents one of the most significant questions in the study of transgenerational effects[26]. Although the vulnerability of germ cell small RNAs to environmental signals is widely realized[27], the specific regulatory pathways involved in transducing environmental cue to germ cell small RNAs are largely unknown. The expression of miRNAs can be regulated through various mechanisms in response to environmental cues such as transcriptional control[28], and the regulation of primary (pri-) or precursor (pre-) miRNAs processing[29,30]. Notably, the transport of pre-miRNA from the nucleus to the cytoplasm also can be controlled by environmental stimuli[31,32]. In locusts, the differences in egg production between solitarious and gregarious locusts are regulated by differential oosorption via miRNA-34 targeting *activin β*[17]. Moreover, the highly expressed miRNA miR-276 in ovaries and eggs of gregarious locusts controls the hatching synchrony of progeny eggs. By recognizing the stem–loop structure of a transcription coactivator gene, *Brahma* (*Brm*), miR-276 upregulates the translation of BRM[16]. BRM is essential for early embryo development and for the stabilization of nucleosome organization[33,34]. Hence, high expression of BRM in locust groups contributes to the developmental homeostasis of embryos, ultimately minimizing the variation in egg-hatching times of gregarious locusts. Thus, gregarious female locusts control the synchronized hatching of progeny eggs through miR-276 targeting BRM, paving the way for large swarm formation and long-distance flight. While miR-276–BRM has been established as a crucial carrier of transgenerational effects, the upstream pathway connecting high density to the production of miR-276 in the reproductive system remains elusive. Deciphering the mechanism through which germline miRNAs respond to high density is critical for unraveling the initial step of transgenerational effects.

In this study, we find that in high-density conditions, FOXN1, a transcription factor (TF) activates an RNA-binding protein PTBP1, which in turn can aid in the nucleo-cytoplasmic transport of pre-miR-276. The enhanced nuclear export of pre-miR-276 leads to high expression of miR-276 in oocytes and hatching synchrony of progeny eggs in gregarious locusts. We reveal that a previously unreported upstream pathway in miRNA biogenesis in female oocytes serves as a potential mechanism for environmental signal-induced transgenerational effects.

## Results
### High density promotes the expression of miR-276 in terminal oocytes of *Locusta migratoria*
Given that miR-276 is expressed at higher levels in the ovaries of gregarious locusts compared to solitarious locusts[16], we aimed to determine the influence of population density on miR-276 expression in terminal oocytes that ultimately give rise to eggs. We assessed miR-276 expression in terminal oocytes of gregarious and solitarious locusts at the same developmental stage (~4.5 mm in length) using specific primers (Fig. 1a, b). Our results revealed that miR-276 expression was nearly 40% higher in the terminal oocytes of gregarious locusts relative to solitarious locusts (Fig. 1b). Thus, a crowded environment can induce the expression of miR-276 in terminal oocytes.

### The elevated nuclear export of pre-miR-276 results in increased expression of miR-276 in terminal oocytes of gregarious locusts
To explore the mechanism by which high population density induces the expression of miR-276 in terminal oocytes, we examined the expression of both transcriptional (pri-miRNA) and posttranscriptional (pre-miRNA) forms of miR-276. We first amplified partial pri-miR-276 sequences (~400 bp) that included pre-miR-276 (Fig. 1c) based on the locust genome[35]. To confirm that the pri-miR-276 sequence containing pre-miR-276 can be properly processed into mature miR-276 in animal cells, we mimicked miR-276 biogenesis in HEK 293T cells, which lack endogenous miR-276 according to the miRbase database (https://www.mirbase.org/). We transfected an expression vector fused with the partial pri-miR-276 sequences (pri-miR-276), or the empty vector (NC) into the cells and determined the expression of mature miR-276. The expression of mature miR-276 was on average 15 times higher in cells transfected with pri-miR-276 compared to NC or no transfection control (blank) (Fig. 1d). The successful expression of mature miR-276 confirmed the accuracy of the pri-miR-276 and pre-miR-276 sequences.

Next, we determined the expressions of pri-miR-276 and pre-miR-276 in the terminal oocytes of gregarious and solitarious locusts, by using specific primers (Fig. 1e and 1f). We observed no significant difference in pri-miR-276 expression levels between gregarious and solitarious locusts (Fig. 1e). Therefore, the transcription of pri-miR-276 is not affected by high density condition. We subsequently examined the posttranscriptional process of miR-276, wherein specific amplification of pre-miR-276 and mature miR-276 (Fig. 1b, f) revealed that the miR-276/pre-miR-276 ratio was 31% higher in gregarious locusts than in solitarious locusts (Fig. 1g), although pre-miR-276 levels were not significantly different (Fig. 1f). Thus, high population density boosts the pre-miR-276 to mature miR-276 processing efficiency. Given that pre-miRNA nuclear export is the key rate-limiting step in miRNA biogenesis[36], we examined the subcellular distribution of pre-miR-276 in the nucleus and cytoplasm of locust terminal oocytes. We measured the ratio of cytoplasmic to nuclear pre-miR-276 to reflect the nuclear export efficiency[37]. The results showed this ratio was nine-fold higher in terminal oocytes of gregarious locusts than in those of solitarious locusts (Fig. 1h). Additionally, fluorescence in situ hybridization (FISH) analysis, utilizing a pre-miR-276 probe (Supplementary Fig. 1), revealed that pre-miR-276 was localized at the periphery of the nucleus, as well as in the cytoplasm of the oocyte (Fig. 1i and Supplementary Fig. 2). Consistent with the qPCR results, the ratio of cytoplasmic to nuclear fluorescence intensity of pre-miR-276 was 15% higher in oocytes of gregarious locusts than in those of solitarious locusts (Fig. 1j). Thus, more pre-miR-276 is exported into the cytoplasm of terminal oocytes in gregarious locusts than in solitarious locusts, leading to increased expression of mature miR-276 in the former.

### PTBP1 promotes the nuclear export of pre-miR-276
To investigate the potential factors responsible for the increased nuclear export efficiency of pre-miR-276, we performed an RNA pull-down assay in locust terminal oocyte extracts. Biotin-labeled RNA probes for the pre-miR-276 sequence were used to identify proteins that interact with pre-miR-276, while a scrambled pre-miRNA from *Caenorhabditis elegans* (*C. elegans*) served as a negative control (NC). Mass spectrometry (MS) analysis identified six protein bands containing 12 proteins that specifically bound with the pre-miR-276 probe (Fig. 2a, Supplementary Fig. 3, and Supplementary Table 1). Protein domain enrichment analysis of these 12 proteins revealed that the nucleotide-binding alpha-beta plait domain superfamily and RNA recognition motif domain were the most enriched proteins, including PTBP1, CELF1, ELAVL3, CPEB1, and hnRNP M (Fig. 2b). To investigate the role of the five proteins in the nuclear export of pre-miR-276, we individually knocked down the genes by injecting female locusts with dsRNAs against each gene at the dorsal site near the adult female ovaries by using Nanoliter Injector, and determined the RNAi efficiency

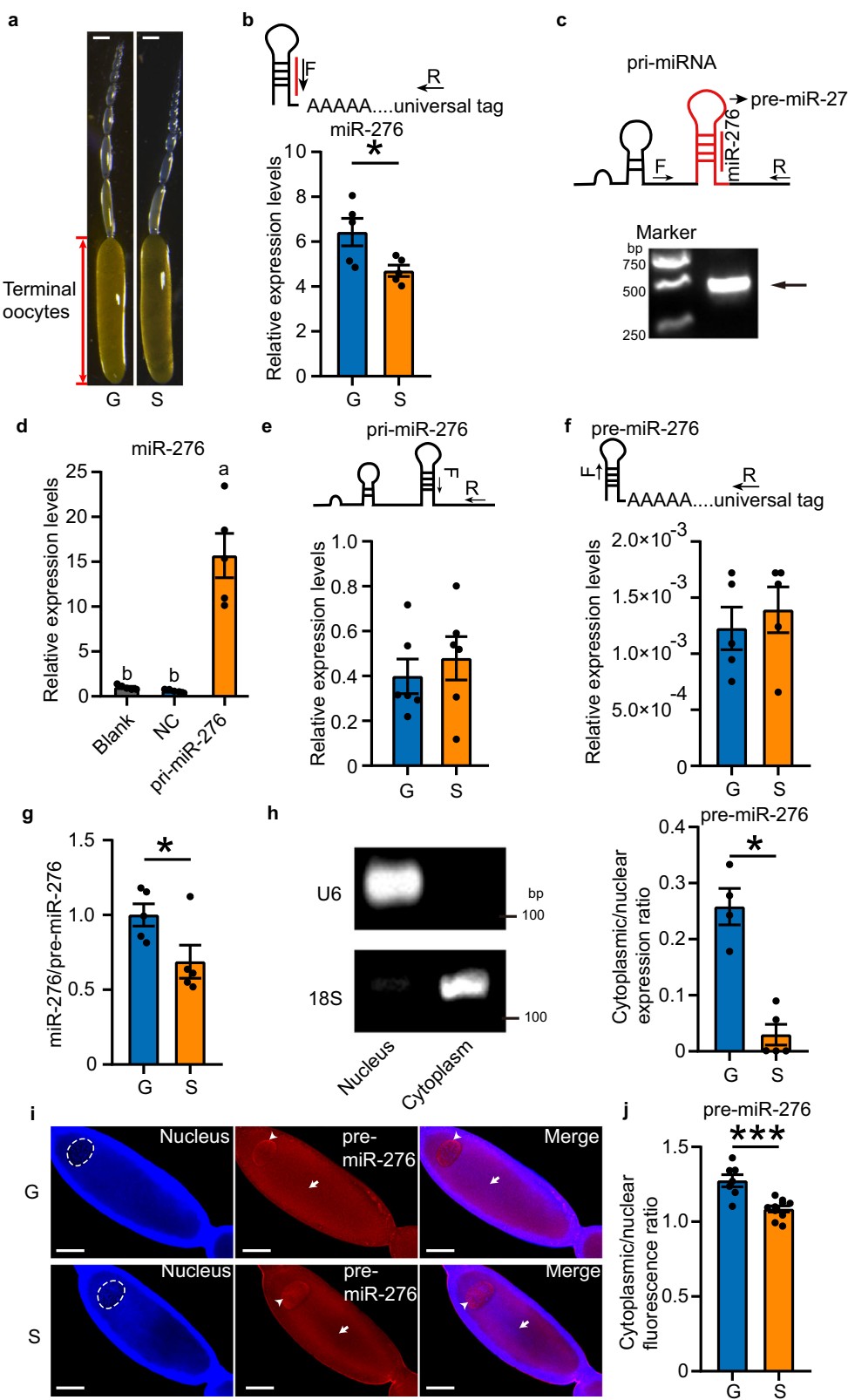

in the terminal oocytes (Supplementary Fig. 4a). *Elavl3* knockdown resulted in decreased expression of pri-miR-276, while knockdown of the other genes did not affect the levels of either pri-miR-276 or pre-miR-276 (Fig. 2c, d). Intriguingly, only *Ptbp1* knockdown caused a reduction in mature miR-276 and miR-276/pre-miR-276 ratio by 22% and 28%, respectively, whereas the other four genes did not have similar effects (Fig. 2e, f). Thus, *Ptbp1* is involved in the maturation of

miR-276 from pre-miR-276 probably through its influence on nuclear export of pre-miR-276.

To confirm the role of PTBP1 in the nuclear export of pre-miR-276, we conducted both in vivo and in vitro experiments. Knockdown of *Ptbp1* in female locusts reduced the ratio of cytoplasmic to nuclear pre-miR-276 by 82%, as determined by qPCR (Fig. 2g and Supplementary Fig. 5). Furthermore, the cytoplasmic to nuclear fluorescence ratio of

**Fig. 1 | The nuclear export efficiency of precursor miR-276 (pre-miR-276) in terminal oocytes of gregarious locusts (G) is higher than solitarious locusts (S). a** Ovarioles from G and S (scale bar: 0.5 mm). **b-c** Amplification of miR-276 (**b**; $n = 5$ biologically independent locusts) and primary miR-276 (pri-miR-276) (**c**). The experiment in (**c**) (bottom) was performed twice, and the arrow indicates the amplicon band of partial pri-miR-276. **d** Expression of miR-276 in HEK 293T cells transfected with pcDNA-pri-miR-276 (pri-miR-276) ($n = 5$ biologically independent cell samples). NC: cells transfected with empty vectors. **e-g** Detection of pri-miR-276 (**e**), pre-miR-276 (**f**), and miR-276/pre-miR-276 ratios (**g**) in terminal oocytes ($n = 6$, 5, and 5 biologically independent locusts for **e** and **f**, and **g**, respectively). **h-j** Nuclear export efficiency of pre-miR-276 in locust terminal oocytes, determined by qPCR (**h**; $n = 4$ and 5 biologically independent locusts for G and S, respectively) and FISH assay (**i** and **j**; For **j**, $n = 7$ and 10 biologically independent locusts for G and S, respectively). U6 and 18S were used as nuclear and cytoplasmic markers,

respectively (**h**). The localization of pre-miR-276 in nucleus and cytoplasm is indicated by arrowhead and arrow, respectively. The nucleus of the terminal oocytes is delineated by a white dotted line circle. Scale bar: 100 μm (**i**). In the diagram of the amplification primers for miR-276 (**b**) and pri-miR-276 (**c**), locations of pre-miR-276 and miR-276 are labeled in red. F forward primer; R reverse primer. Statistically significant differences between groups in (**d**) are shown by different letters using one-way ANOVA followed by Tukey's post hoc tests with two-stage step-up method of Benjamini, Krieger and Yekutieli adjustment for multigroup comparisons. Mann–Whitney $U$ test (two-tailed) was applied for two-group comparisons in (**g**) and (**h**), and Student's $t$ test (two-tailed) was applied for other two-group comparisons. The data are shown as the mean ± SEM. * $P < 0.05$, *** $P < 0.001$. The original scans for Figs. 1c and 1h are provided in Supplementary Fig. 15a and 15b, respectively. Source data and details of the statistical results are provided as a Source Data file.

pre-miR-276 was reduced by 16% upon knockdown of *Ptbp1*, as shown by FISH assay (Fig. 2h). Additionally, we co-transfected HEK 293T cells with recombinant vectors containing the *Ptbp1* gene (PTBP1) and pri-miR-276. Constructs fused with the *LacZ* gene (LacZ) and empty vectors of pcDNA (NC) served as controls for PTBP1 and pri-miR-276, respectively. Compared to LacZ control, expression of locust PTBP1 significantly increased the miR-276/pre-miR-276 ratio by two folds (Fig. 2i). Finally, we observed a 39% increase in the cytoplasmic to nuclear ratio of pre-miR-276 abundance upon PTBP1 transfection (Fig. 2j), further supporting the role of PTBP1 in promoting nuclear export of pre-miR-276. Collectively, our findings demonstrated that PTBP1, an RNA-binding protein[38], can enhance the nuclear export of pre-miR-276 in terminal oocytes of gregarious locusts.

## High density activates the transcription of *Ptbp1* by FOXN1

As an RNA binding protein, PTBP1 typically serves as a downstream effector of signal transduction[39], leading us to speculate that other upstream regulators might act on PTBP1 in response to high density. Then we set out to identify the transcription factor (TF) responsible for regulating *Ptbp1*, because of higher *Ptbp1* transcription level under high-density conditions (Fig. 3a). Firstly, we retrieved an approximately 1.8 kb upstream DNA sequence of the ATG start codon of *Ptbp1* gene based on locust genome[35] (Fig. 3b). We then progressively truncated upstream sequences to evaluate the promoter activity (Fig. 3b and 3c). The transfection of all these sequences fused with pGL4.10 plasmid showed higher promoter activity than the empty vector (NC), revealed by the luciferase activity assay (Fig. 3d). Notably, the sequence 2 displayed a substantial increase of promoter activity, over 15 times relative to NC (Fig. 3d). The results indicate the region between -1391 bp to -823 bp upstream of *Ptbp1* may represent the binding site of critical TFs. Hence, we predicted the potential TFs that can bind to the -1391 bp to -823 bp upstream sequences by using MatInspector program (http://www.genomatix.de/matinspector.html)[40], to uncover the key factor that is crucial for *Ptbp1* transcription activation (see Materials and methods). We identified four TFs including Onecut, Paired (Prd), Antennapedia (Antp), and Forkhead box protein N1 (FOXN1) as potential candidates (Fig. 3e). In addition, we predicted the potential TFs based on the gene-TF co-expression principle[41], and identified E(spl)mγ-HLH (Mγ) as a candidate TF that may contribute to *Ptbp1* transcriptional activation.

In order to screen the key TF for *Ptbp1*, we determined the expression patterns of the five TFs in various tissues of locusts. Among these five TFs, the mRNA of *Onecut* and *Antp* were not expressed in the terminal oocytes of locusts (Fig. 3f). Thus, we knocked down the other three TFs one by one to verify their roles in *Ptbp1* transcriptional activation. We injected the respective dsRNAs at the dorsal site near the adult female ovaries by using Nanoliter Injector in order to deliver the dsRNAs to the ovaries as locally as possible. Then we determined the RNAi efficiency in the terminal oocytes (Supplementary Fig. 4b). Knockdown of *Mγ* did not affect the expression of *Ptbp1* and the

production of miR-276. Knockdown of *Prd* also did not alter the production of miR-276, although resulted in a slight increase of *Ptbp1* ($P = 0.18$). However, knockdown of *Foxn1* led to a reduction in the expression of *Ptbp1* and the production of miR-276 by 41% and 27%, respectively (Fig. 3g and 3h). Furthermore, the expression of both mRNA and protein of *Prd* in the terminal oocytes were not significantly different between gregarious and solitarious locusts (Supplementary Fig. 6). Thus, Prd may not be the transcription factor of *Ptbp1*. Notably, in line with the higher expression levels of *Ptbp1* and miR-276 induced by high population density, the protein level of FOXN1 in terminal oocytes of gregarious locusts also was three-fold higher than solitarious locusts, although the mRNA expression of *Foxn1* was not affected by population density (Fig. 3i, j). Thus, these results suggest that high expression of FOXN1 in crowding stimuli promotes the expression of *Ptbp1*.

To further validate the activation effects of FOXN1 on transcription of *Ptbp1*, we firstly determined the binding of FOXN1 with the promoter of *Ptbp1*. We conducted electrophoretic mobility shift assay (EMSA) on terminal oocytes employing a biotin-labeled probes (WT probe) targeting the predicted binding site of FOXN1 (Fig. 3k). The WT probe exhibited binding ability with nuclear proteins, resulting in a shift band. Furthermore, competition with an unlabeled probe decreased the intensity of the shift band, whereas no competitive effect was observed with the unlabeled mutated probe (Mut probe) (Fig. 3l). Additionally, when employing biotin-labeled mutant probe without WT probe, the shift band was disappeared (Supplementary Fig. 7). Furthermore, the intensity of the shifted band was reduced by using of anti-FOXN1 antibody, and by knocking down of *Foxn1* (Fig. 3m). These results confirmed the binding of FOXN1 with the promoter of *Ptbp1* in terminal oocytes of locusts. Subsequently, we proceeded to test the capacity of FOXN1 to affect the promoter activity of sequence 2, comprising the putative FOXN1 binding site, in *Drosophila Schneider* (S2) cells. In order to eliminate the probable interference from endogenous *Foxn1* gene named *Jumu* in *Drosophila*, which shares 78% protein sequence identity with locust FOXN1, we first knocked down *Jumu* in S2 cells (Supplementary Fig. 4c). Then, by overexpressing locust FOXN1, we observed a significant five-fold boost in promoter activity (Fig. 3n), thereby confirming crucial role played by FOXN1 as a transcription factor of *Ptbp1*.

## PTBP1 facilitates the nuclear export of pre-miR-276 by interacting with XPO5

We sought to explore the mechanism by which PTBP1 promotes the nuclear export of pre-miR-276 in locusts. We first confirmed the direct interaction between PTBP1 and pre-miR-276 by using immunoblotting followed with RNA pull-down, and RNA immunoprecipitation (RIP) assays. We detected the presence of PTBP1 in the complex pulled down by pre-miR-276 from the lysates of terminal oocytes, as well as the enrichment of pre-miR-276 in the RNA immunoprecipitates of PTBP1 (Fig. 4a, b). Furthermore, we performed a pull-down assay using prokaryotically expressed PTBP1 (Supplementary Fig. 8), confirming the

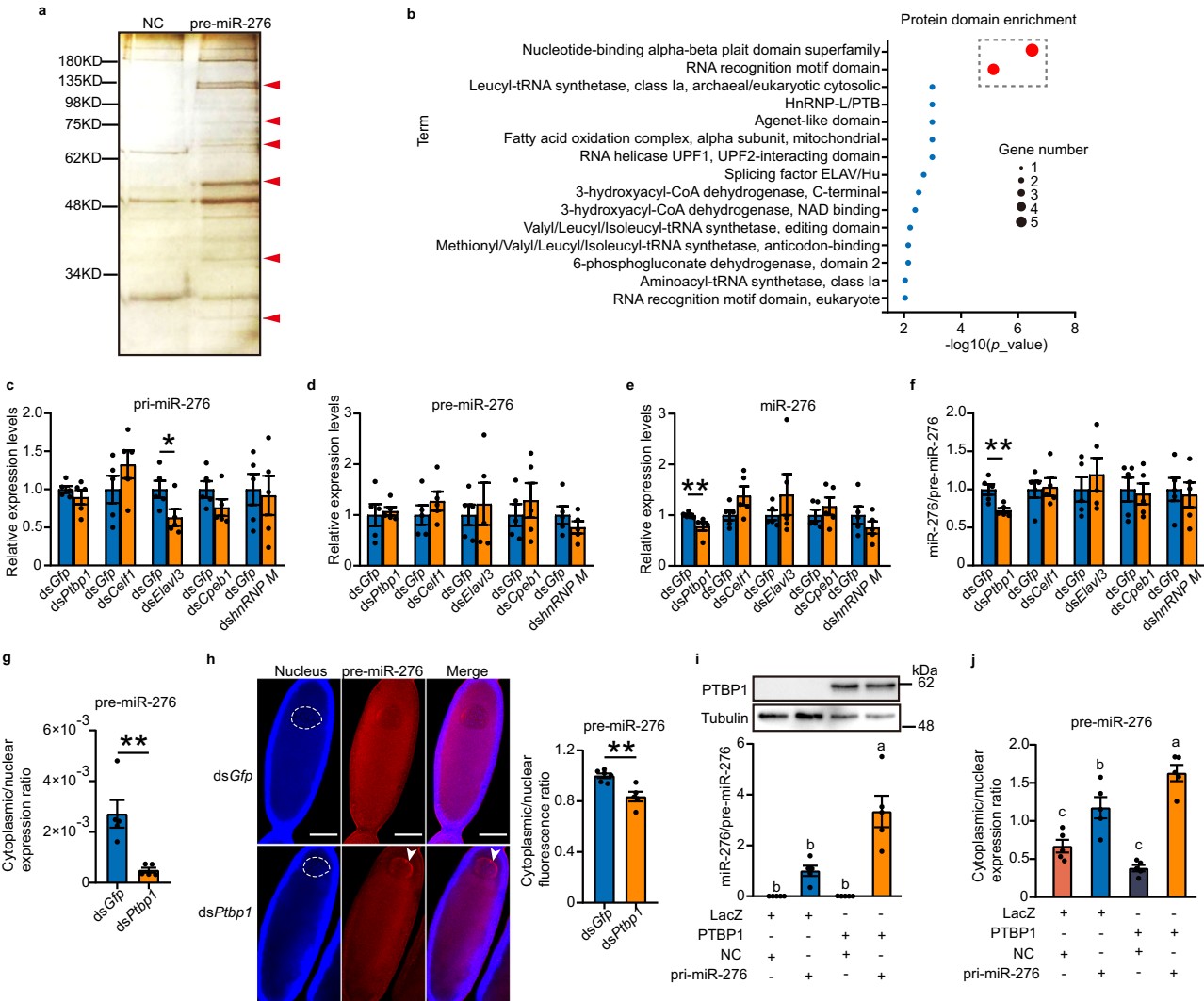

**Fig. 2 | PTBP1 facilitates the nuclear export of precursor miR-276 (pre-miR-276).** **a** RNA pull-down assay of pre-miR-276 in locust terminal oocytes. The protein bands specifically bound with pre-miR-276 are indicated by red arrowheads. The experiment was performed one time with two biologically independent locust samples. **b** Domain enrichment assay following MS of the proteins from (a). Protein domain enrichment methods were implemented in the LocustMine platform[41]. The terms of protein domain with P < 0.01 were regarded as enriched. **c-f** Effects of knockdown of the five candidate genes on the expression levels of primary miR-276 (pri-miR-276) (**c**), pre-miR-276 (**d**), miR-276 (**e**) and miR-276/pre-miR-276 ratio (**f**) (n = 5 biologically independent locusts). **g-h** Effects of *Ptbp1* knockdown on nuclear export of pre-miR-276, determined by qPCR (**g**; n = 5 biologically independent locusts) and FISH (**h**; n = 6 and 5 biologically independent locusts for ds*Gfp* and ds*Ptbp1*, respectively). The nucleus of the terminal oocytes is delineated by a white dotted line circle. Arrowheads indicate the locations of pre-miR-276 in the nucleus.

Scale bar: 100 μm. **i-j** Effects of PTBP1 overexpression on miR-276/pre-miR-276 ratios (**i**) and nuclear export of pre-miR-276 (**j**) in HEK 293T cells (n = 5 biologically independent cell samples). LacZ LacZ-fused vector; PTBP1 PTBP1-fused vector; NC pcDNA vector; pri-miR-276 pcDNA fused with pri-miR-276. Statistically significant differences between groups in (**i**) and (**j**) are shown by different letters using one-way ANOVA followed by Tukey's post hoc tests with two-stage step-up method of Benjamini, Krieger and Yekutieli adjustment for multigroup comparisons. Mann–Whitney U test (two-tailed) was applied for two-group comparisons of ds*Ptbp1* and ds*Cpeb1* in (**e**), ds*Ptbp1* in (**g**), and Student's t test (two-tailed) was used for other two-group comparisons. The original scans for Fig. 2a and Fig. 2i are provided in Supplementary Figs. 3 and 17a, respectively. The data are shown as the mean ± SEM. * P < 0.05, ** P < 0.01. Source data and details of the statistical results are provided as a Source Data file.

direct interaction between PTBP1 and pre-miR-276 (Fig. 4c). Therefore, PTBP1 may facilitate the nuclear export of pre-miR-276 by directly interacting with it in the terminal oocytes of locusts.

To further examine the role of PTBP1 in the nuclear export of pre-miRNA, we investigated whether it functions as an exportin protein. Given that transporters can pass through nuclear pore channels[42], we tested whether PTBP1 is a nucleus-cytoplasm shuttling protein. Indeed, we detected the presence of PTBP1 in both the nucleus and cytoplasm of locust terminal oocytes by immunoblotting and immunostaining (Fig. 4d, e). Additionally, we attempted to determine whether PTBP1 can export the pre-miR-276 from nucleus to cytoplasm independently of XPO5, the primary exportin protein for most pre-miRNAs[43]. We

utilized HEK 293T cells again to prevent interference from endogenous miR-276. Deletion of XPO5 protein in HEK 293T cells (Fig. 4f and Supplementary Fig. 9), resulted in a significant nuclear retention of pre-miR-276 (Fig. 4g). Furthermore, overexpression of locust PTBP1 in XPO5-deleted cells did not relieve the blockade of nuclear export of pre-miR-276 (Fig. 4g). These data indicate that while PTBP1 can function as an exportin protein for pre-miR-276, it cannot substitute for the function of XPO5 in nuclear transport.

Our results suggest that PTBP1 may facilitate the nuclear export of pre-miR-276 by recruiting XPO5. To investigate this further, we used algorithm prediction and co-immunoprecipitation (co-IP) assays to examine the potential interaction between PTBP1 and XPO5. Using

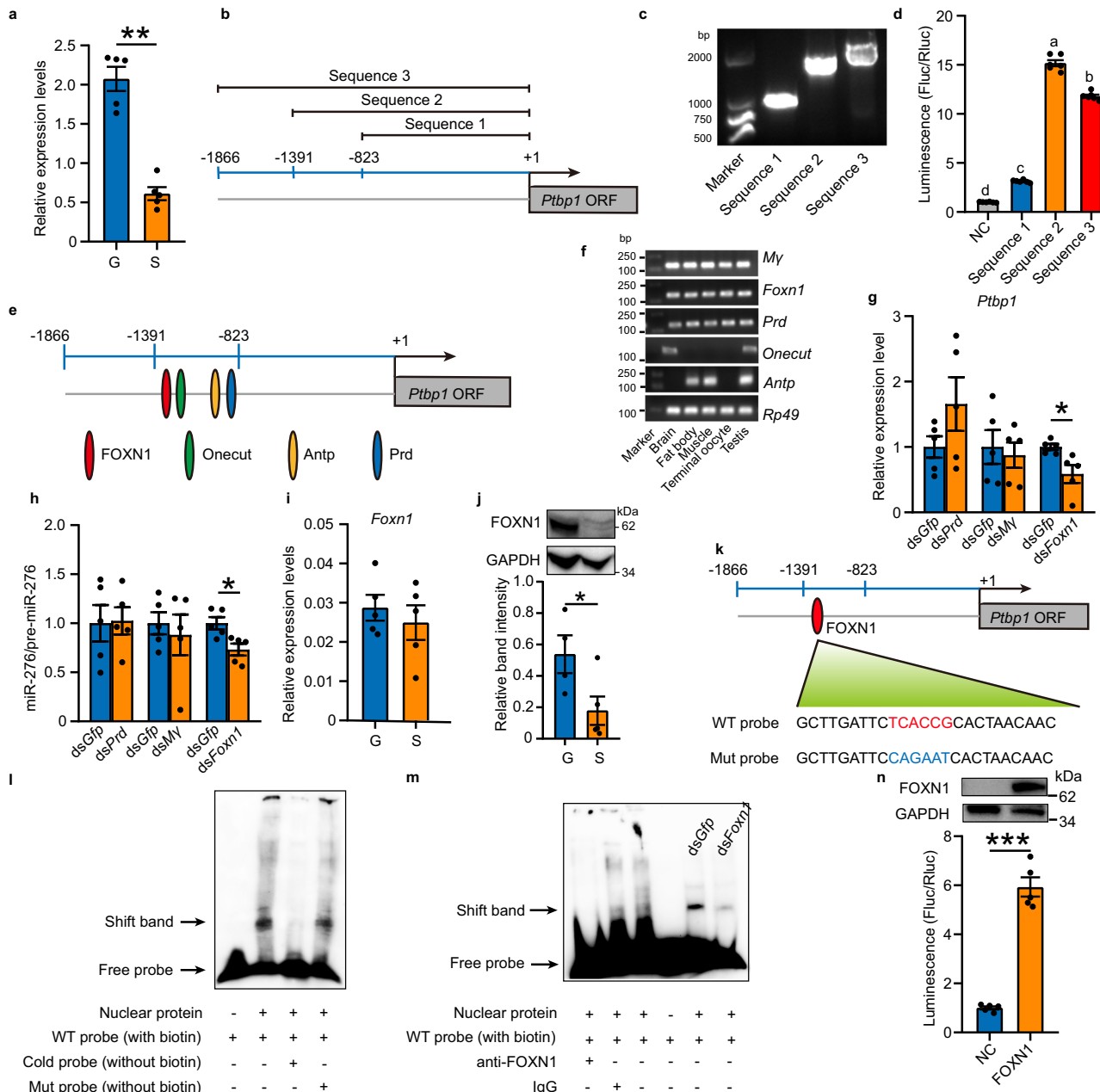

**Fig. 3 | *Ptbp1* is transcriptionally activated by FOXN1. a** Expression of *Ptbp1* in terminal oocytes of gregarious (G) and solitarious (S) locusts (*n* = 5 biologically independent locusts). **b-d** A diagram (**b**), PCR results (**c**), and promoter activity (**d**) of the progressively truncated upstream sequences of ATG start codon of *Ptbp1* (*n* = 6 biologically independent cell samples). **e** A diagram for binding of the predicted transcription factors (TFs) on *Ptbp1* promoter. **f** PCR of the TFs candidates from various tissues. **g-h** The effects of three TFs knockdown on the expression of *Ptbp1* (**g**) and miR-276/pre-miR-276 ratio (**h**) (*n* = 5 biologically independent locusts). **i-j** Expression of mRNA (**i**; *n* = 5 biologically independent locusts) and protein (**j**; *n* = 4 and five biologically independent locusts respectively for G and S) of *Foxn1*. **k-m** Validation of the binding of FOXN1 with the *Ptbp1* promoter by EMSA from the nuclear extracts of terminal oocytes. Red and blue letters indicate the binding sites of wild-type (WT) probe and the mutations in mutated (Mut) probe, respectively (**k**).

The bands of free probe and shift band are indicated by arrows (**l** and **m**). **n** The effects of overexpression of locust FOXN1 (top) on the *Ptbp1* promoter determined by dual-luciferase assay (bottom) (*n* = 5 biologically independent cell samples). *Rp49* (**f**) and GAPDH (**j** and **n**) were used as internal control for genes and proteins, respectively. Statistically significant differences between groups in (**d**) are shown by different letters using one-way ANOVA followed by Tukey's post hoc tests with two-stage step-up method of Benjamini, Krieger and Yekutieli adjustment for multigroup comparisons. Mann–Whitney *U* (two-tailed) test was applied for two-group comparisons in (**a**), and Student's *t* test (two-tailed) was applied for other two-group comparisons. NC cells transfected with empty vector. The data are shown as the mean ± SEM. * *P* < 0.05, ** *P* < 0.01, *** *P* < 0.001. The original scans for Fig. 3c, 3f, 3j, and 3n are provided in Supplementary Figs. 15a, 15c, 17b and 17c, respectively. Source data and details of the statistical results are provided as a Source Data file.

InterEvDock2[44–47], we predicted that locust PTBP1 can bind with XPO5 (Fig. 4h). We further performed a co-IP assay in locust terminal oocytes, showing that PTBP1 can be immunoprecipitated by the antibody against XPO5, and vice versa (Fig. 4i). To determine if PTBP1 promotes the binding of pre-miR-276 to XPO5 to aid the nuclear export of pre-miR-276, we performed RIP assay of XPO5 in locust terminal oocytes after knockdown of *Ptbp1*. The results showed that silencing *Ptbp1* reduced the binding of pre-miR-276 to XPO5 by 39% (Fig. 4j). In addition, we tested whether overexpression of locust PTBP1 in HEK 293T cells could enhance the binding of pre-miR-276 to XPO5. The association of human XPO5 and locust PTBP1 was predicted by Inter-EvDock2 (Fig. 4k), and confirmed with a co-IP assay in HEK 293T cells

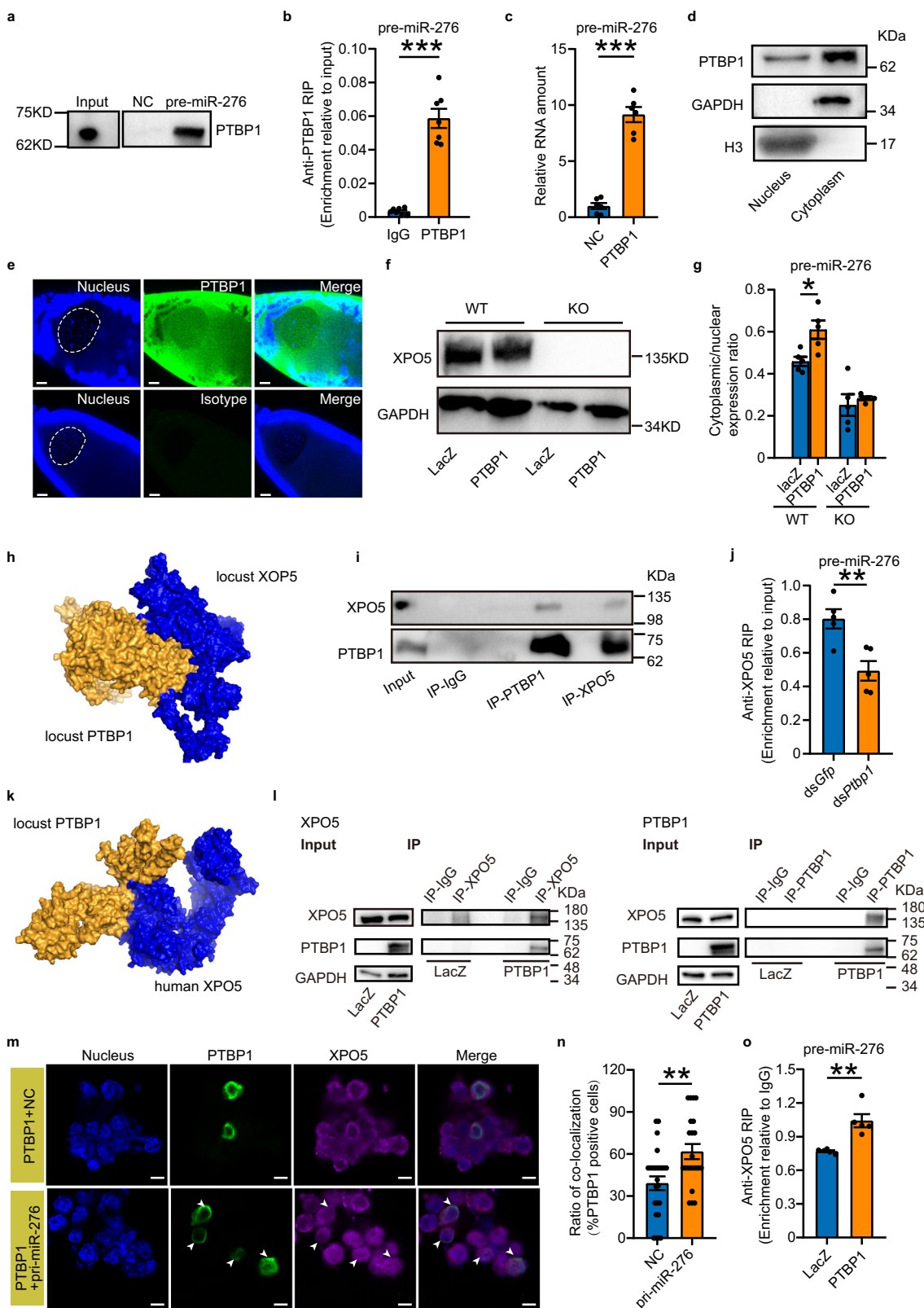

(Fig. 4l). We further investigated the co-localization between PTBP1 and XPO5, as well as the effect of PTBP1 overexpression on the binding of pre-miR-276 to XPO5. We observed a 58% increase in co-localization of PTBP1 and XPO5 upon transfection of pri-miR-276 (Fig. 4m, n). Importantly, the binding of pre-miR-276 to XPO5 was increased by 35% upon overexpression of PTBP1 (Fig. 4o). Collectively, the findings indicate that PTBP1 promotes the transport of pre-miR-276 from the nucleus to the cytoplasm by enhancing the binding of XPO5 with pre-miR-276.

## PTBP1 facilitates the nuclear export of pre-miR-276 by binding with the "CU motifs"

To investigate whether PTBP1 can pervasively promote the nuclear export of pre-miRNAs, we assessed the effects of *Ptbp1* on the

**Fig. 4 | PTBP1 coupled with XPO5 facilitates the nuclear export of precursor miR-276 (pre-miR-276). a** Immunoblotting of PTBP1 following RNA pull-down of pre-miR-276. **b-c** qPCR analysis of pre-miR-276 in PTBP1 immunoprecipitates from terminal oocyte lysates (**b**; $n = 7$ biologically independent locusts) and *E.coli* (**c**; $n = 6$ biologically independent *E.coli* samples). **d-e** Nuclear/cytoplasmic distribution of PTBP1 in terminal oocytes. H3 and GAPDH on the same gel were used as markers for nuclear and cytoplasmic proteins, respectively. The nucleus of the terminal oocytes is delineated by a white dotted line circle. Scale bar: 20 μm. **f-g** Expression of XPO5 (**f**) and cytoplasmic/nuclear ratio of pre-miR-276 (**g**) in wild-type (WT) and XPO5-deleted (KO) HEK 293T cells co-transfected pcDNA-pri-miR-276 (pri-miR-276) with PTBP1 or LacZ ($n = 5$ biologically independent cell samples). **h** Predicted interaction between locust PTBP1 and XPO5. **i** Co-IP of XPO5 and PTBP1 in terminal oocytes. **j** RIP assay using XPO5 antibody in terminal oocytes after *Ptbp1* knockdown ($n = 5$ biologically independent locusts). **k** Predicted interaction between

locust PTBP1 and human XPO5. **l** Co-IP of PTBP1 and XPO5 in HEK 293T cells co-transfected pri-miR-276 with LacZ or PTBP1. **m** Co-localization of PTBP1 and XPO5 in HEK 293T cells co-transfected with PTBP1 and pri-miR-276 or empty vector (NC). Arrowheads indicate the locations of pre-miR-276. Scale bar: 10 μm. **n** The ratio of co-localization locust PTBP1 and XPO5 in PTBP1-positive cells ($n = 25$ and 22 independent microscope visions for NC group and pri-miR-276 group, respectively). **o** RIP assay using XPO5 antibody in HEK 293T cells co-transfected with pri-miR-276 and LacZ or PTBP1 ($n = 5$ biologically independent cell samples). GAPDH was used as internal control in (**f**) and (**l**). Mann–Whitney $U$ test (two-tailed) was applied for two-group comparisons in (**n**), Student's $t$ test (two-tailed) was applied for other two-group comparisons. The data are shown as the mean ± SEM. * $P < 0.05$, ** $P < 0.01$, *** $P < 0.001$. The original scans for Fig. 4a, 4d, 4f, 4i, and 4l are provided in Supplementary Fig. 17d, 17e, 17f, 17g and 17h, respectively. Source data and details of the statistical results are provided as a Source Data file.

subcellular localization of eight miRNAs. These miRNAs were selected as they are expressed at higher levels in the ovaries of gregarious locusts when compared to solitarious locusts[16]. Our results revealed that aside from the enhanced nuclear export efficiency of pre-miR-305 and pre-miR-34, the nuclear export of the remaining six pre-miRNAs was unaffected by *Ptbp1* knockdown (Fig. 5a). In accordance with this, the binding of pre-miR-305 and pre-miR-34 with XPO5 was augmented by *Ptbp1* knockdown, while the binding of other six pre-miRNAs with XPO5 remained unchanged (Supplementary Fig. 10a). Nevertheless, all these pre-miRNAs were unable to bind with PTBP1 (Supplementary Fig. 10b). Thus, the increased export efficiency caused by *Ptbp1* knockdown may be associated with some indirect factors. These findings suggest that PTBP1 may promote the nuclear export of pre-miR-276 by recognizing a specific sequence motif, instead of the stem-loop structure of pre-miRNAs.

To investigate key motifs recognized by PTBP1, we analyzed the sequence of pre-miR-276 and identified two "CU motifs" on the 5p and 3p strands (Fig. 5b) (see Methods for identification of "CU motifs"). These findings are in line with PTBP1's character as a polypyrimidine tract-binding protein[38]. To determine whether the "CU motifs" are responsible for PTBP1-mediated nuclear export of pre-miR-276, we conducted both single and double mutation assays in HEK 293T cells (Fig. 5b). None of the mutation constructs affected the predicted stem-loop structure of pre-miRNAs (Fig. 5c). Apart from the double mutation of "CU motifs" (Mut 4) and the "CU motif" on the 5p strand in conjunction with the "non-CU motif" (Mut 5), all other mutations were able to produce mature miRNAs (Supplementary Fig. 11). Nevertheless, all the mutation vectors can generate pre-miRNAs (Fig. 5d), enabling us to detect the nuclear export of pre-miRNAs. The enhancing effects of PTBP1 on nucleocytoplasmic transport of pre-miRNAs and the subsequent increase in mature miRNA production were abolished when any one or both of the "CU motifs" were mutated (Fig. 5e–h). Furthermore, a single mutation in the "CU motif" on the 5p or 3p strands, and all of the double mutations failed to bind with PTBP1 (Fig. 5i and 5j). Interestingly, PTBP1's enhancement of XPO5 binding to pre-miRNAs was blocked when any one or both of the "CU motifs" were mutated (Fig. 5k and 5l). However, PTBP1 still can significantly increase both the nuclear export efficiency of pre-miRNA and miRNA production when only the "non-CU motif" was mutated (Fig. 5e and 5g). Additionally, a single mutation of the "non-CU motif" did not impair the pre-miRNA binding to PTBP1 (Fig. 5i), nor did it impact PTBP1's ability to augment XPO5 binding to pre-miRNA (Fig. 5k). Therefore, both "CU motifs" in 5p and 3p strands of pre-miR-276 are required for PTBP1-mediated nuclear export. Additionally, due to the absence of the "CU motif" in the sequences pre-miR-305 and pre-miR-34 (Supplementary Fig. 12), further supporting that the enhanced nuclear export of these two pre-miRNAs by *Ptbp1* knockdown (Fig. 5a) may be attributed to some indirect reasons. Overall, PTBP1 promotes the nuclear export of pre-miR-276 by specifically recognition of "CU motifs".

## PTBP1 promotes egg-hatching synchrony by activating miR-276

Can PTBP1 affect egg-hatching synchrony of gregarious locusts? Our previous study has found that the egg-hatching synchrony of gregarious locusts are controlled by miR-276 in female ovaries via upregulating BRM[16], which is critical for the early embryo development of *Drosophila* and for stability of transcription[33,34]. Additionally, synchronous hatching is indicated by smaller variations in hatching time and the shorter durations of hatching peak (10–90% hatching) in eggs laid by gregarious females than solitarious females[16] (Supplementary Fig. 13a). Furthermore, inhibition and overexpression of miR-276 in gregarious and solitarious females lead to increased and decreased variations of egg-hatching time, respectively[16] (Supplementary Fig. 13b and 13c). We observed that the expression of PTBP1 in terminal oocytes of gregarious locusts was significantly higher than solitarious locusts (Supplementary Fig. 14a). Furthermore, PTBP1, miR-276, and the target of miR-276 BRM were more abundant in eggs of gregarious locusts than in those of solitarious locusts (Supplementary Fig. 14b, 14c and 14d), indicating the transmission of the expression patterns of PTBP1, miR-276, and BRM to the offspring eggs. To further investigate the role of *Ptbp1* in regulating egg-hatching synchrony, we conducted a *Ptbp1* knockdown experiment on female gregarious locusts. The results showed that the levels of PTBP1, miR-276, and BRM were reduced by 68.5%, 72.8%, and 28.5%, respectively, in the progeny eggs upon knockdown of *Ptbp1* in female locusts (Supplementary Fig. 14e, 14f, 14g). The hatching of progeny eggs became more heterochronic, with a significant increase in the variation of egg-hatching time and a longer duration of the egg-hatching peak (10–90% hatching). (Fig. 6a and 6b and Supplementary Fig. 13d). Furthermore, we performed rescue assays by overexpressing miR-276 in gregarious females that were previously injected with ds*Ptbp1* and found that it offset the effects of ds*Ptbp1* on decreased expression of miR-276 and BRM (Supplementary Fig. 14h and Supplementary Fig. 14i), and the hatching asynchrony of progeny eggs (Fig. 6c and 6d, and Supplementary Fig. 13e). Therefore, PTBP1-mediated egg-hatching synchrony functions by promoting the expression of miR-276.

Locusts lay eggs in clusters known as egg pods[48], which are synonymous with egg clutches. The hatching synchrony within egg pods is a crucial factor in ensuring the overall synchrony of the eggs. Moreover, considering the significance of hatching synchrony within egg pods for avoiding cannibalism[49], we attempted to investigate the hatching synchrony within egg pods. After standardizing the hatching time of each egg pod, we found that the egg-hatching within egg pod was more synchronous in gregarious locusts than solitarious locusts (Supplementary Table 2). Inhibition or overexpression of miR-276 in gregarious or solitarious females resulted in increased and decreased variation of hatching time within egg pod (Supplementary Table 2). Moreover, knockdown of *Ptbp1* led to asynchronous hatching, and the effects caused by *Ptbp1* knockdown can be rescued by overexpression of miR-276. The results of hatching synchrony within egg pod were consistent with the entire egg-hatching synchrony.

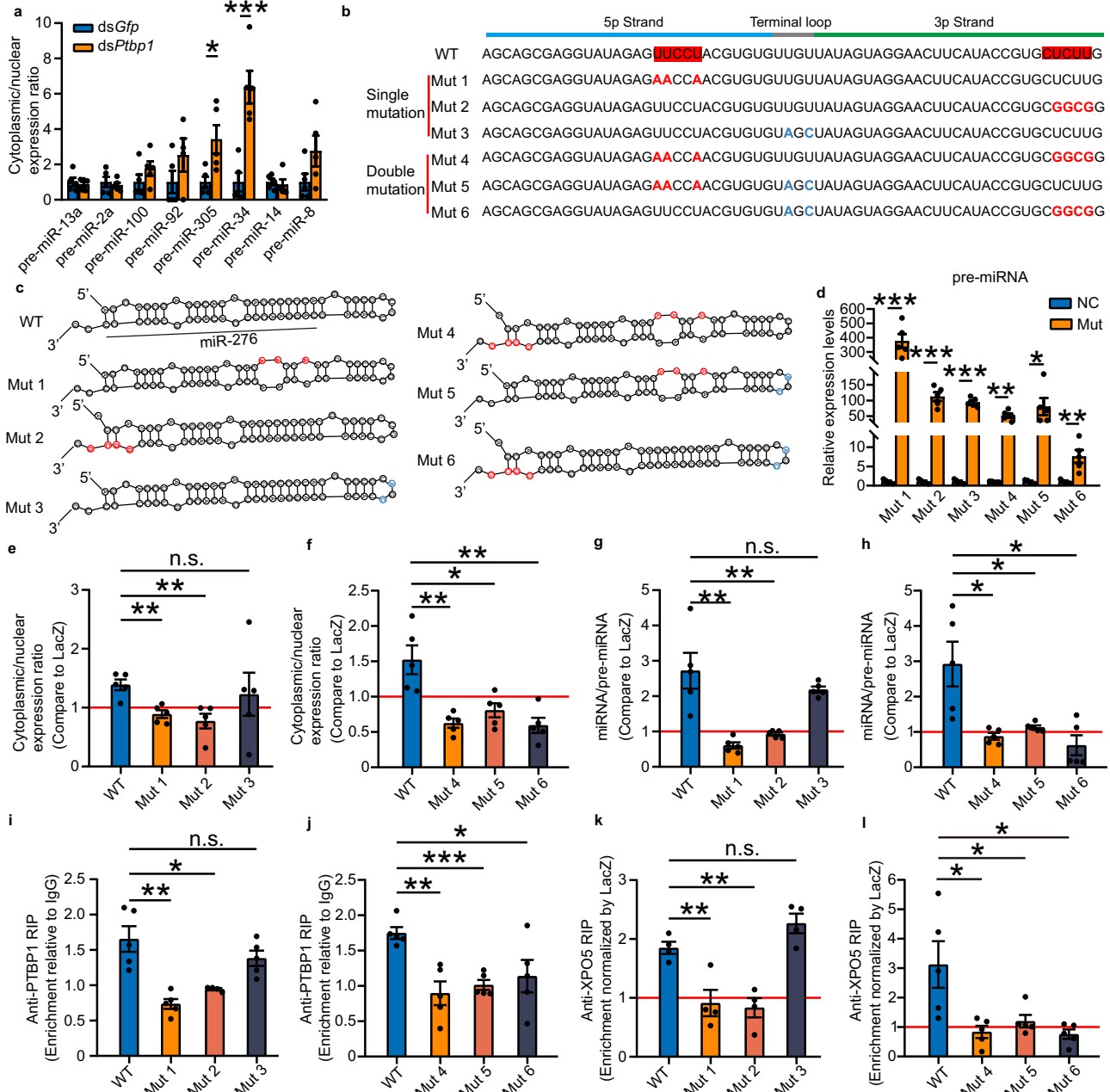

**Fig. 5 | PTBP1-mediated nuclear export is dependent on the "CU motif".**
**a** Cytoplasmic/nuclear ratios of the eight selected pre-miRNAs in terminal oocytes after knockdown of *Ptbp1* (*n* = 5 biologically independent locusts). **b** Sequences of wild type (WT) and mutant pre-miR-276 (Mut). Red boxes indicate the "CU motif" sequences. The red and blue letters indicate the mutant bases within the "CU motif" and "non-CU motif", respectively. **c** Secondary structures of WT and Muts. **d** Expression of pre-miRNAs produced from the mutated plasmids (*n* = 5 biologically independent cell samples). **e-f** Effects of PTBP1 on cytoplasmic/nuclear pre-miRNA ratios in HEK 293T cells. The ratio was normalized to that of the LacZ transfection group (*n* = 5 biologically independent cell samples). The red line indicates that the cells transfected with PTBP1 or LacZ had the same cytoplasmic/nuclear pre-miRNA ratio. **g-h** miRNA/pre-miRNA ratios in cells transfected with the wild-type or mutated plasmids. The ratio was normalized to that of the LacZ transfection group (*n* = 5 biologically independent cell samples). The red line indicates that the cells transfected with PTBP1 or LacZ had the same miRNA/pre-miRNA ratio. RIP assay using antibodies against PTBP1 (**i**–**j**) or XPO5 (**k**–**l**) in HEK 293T cells after transfection with WT and mutated vectors (**i**: *n* = 5 biologically independent cell samples for Mut1 and Mut3 group, *n* = 4 biologically independent cell samples for Mut2 group; **j** and **l**: *n* = 5 biologically independent cell samples; **k** *n* = 4 biologically independent cell samples). The red line indicates that the cells transfected with PTBP1 or LacZ had the same abundances of pre-miRNAs in the immunoprecipitates of XPO5. Mann–Whitney *U* test (two-tailed) was applied for two-group comparisons of pre-miR-13a, pre-miR-100, pre-miR-92 and pre-miR-14 in (**a**), mut 4 in (**d**) and mut 6 in (**h**). Student's *t* test (two-tailed) was applied for other two-group comparisons. The data are shown as the mean ± SEM. * *P* < 0.05, ** *P* < 0.01, *** *P* < 0.001, n.s. no significant difference. Source data and details of the statistical results are provided as a Source Data file.

Overall, our results support that high population density enhances the PTBP1-miR-276 pathway in female oocytes, thereby leading to hatching synchrony of progeny eggs in gregarious locusts.

## PTBP1-mediated nuclear export of pre-miR-276 in *Drosophila*
We next investigate whether PTBP1-mediated nuclear export of pre-miR-276 is an evolutionary conserved mechanism in insects. As miR-276 is an insect-specific miRNA (https://www.mirbase.org/), the pre-

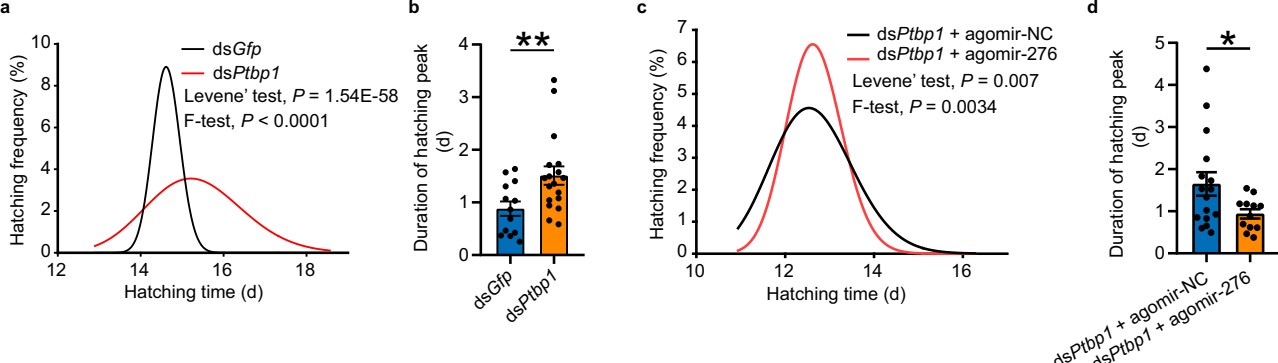

**Fig. 6 | PTBP1–miR-276 regulates egg-hatching synchrony. a-b** Normal curve fitting of hatching time (**a**) and duration of hatching peaks (**b**) of the eggs produced by female locusts injected with ds*Gfp* and ds*Ptbp1* (for normality test: *n* = 1041 and 769 biologically independent eggs for ds*Gfp* and for ds*Ptbp1*, respectively; for comparison of hatching peak during, *n* = 13 and 18 biologically independent egg pods for ds*Gfp* and ds*Ptbp1*, respectively). **c-d** Normal curve fitting of hatching time (**c**) and duration of hatching peaks (**d**) of eggs produced by female locusts injected with agomir-NC or agomir-276 pre-treated with ds*Ptbp1* (for normality test: *n* = 403

and 243 biologically independent eggs for agomir-NC and agomir-276, respectively; for comparison of hatching peak during, *n* = 16 and 12 biologically independent egg pods for agomir-NC and agomir-276 group, respectively). Levene's test (one-tailed) and F-test (one-tailed) were used to analyze the variation of the egg hatching time in (**a**) and (**c**). Mann–Whitney *U* test (one-tailed) was applied for two-group comparisons in (**b**) and (**d**). The data are shown as the mean ± SEM. * *P* < 0.05, ** *P* < 0.01. Source data and details of the statistical results are provided as a Source Data file.

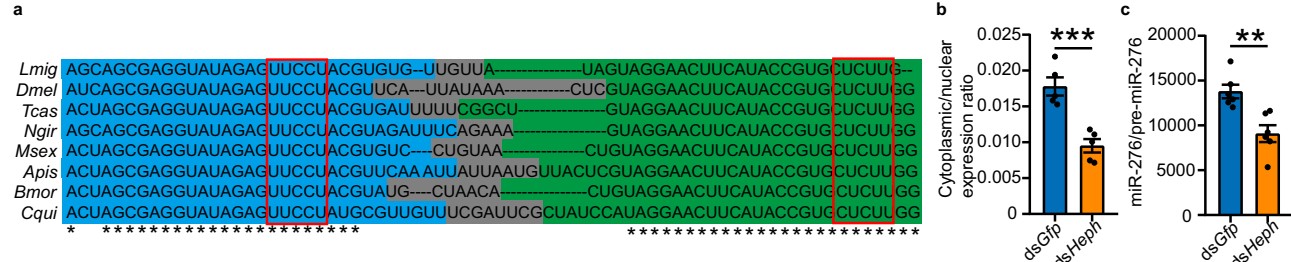

**Fig. 7 | PTBP1 promotes nuclear export of pre-miR-276 in *Drosophila*.**
**a** Alignment of the pre-miR-276 sequence among insect species. *Lmig*: *Locusta migratoria*, *Dmel*: *Drosophila melanogaster*, *Tcas*: *Tribolium castaneum*, *Ngri*: *Nasonia giraulti*, *Msex*: *Manduca sexta*, *Apis*: *Acyrthosiphon pisum*, *Bmor*: *Bombyx mori*, *Cqui*: *Culex quinquefasciatus*. The asterisks indicate the conserved sequence among these insects. The 5p strand, 3p strand, and terminal loop of pre-miR-276 are emphasized by blue, green, and gray areas, respectively. The red boxes show the "CU motifs" in pre-miR-276. **b** Cytoplasmic/nuclear pre-miR-276 ratio in *Drosophila*

S2 cells after *Heph* knockdown (*n* = 5 biologically independent cell samples, each cell sample was from one well of a 48-well plate). **c** miR-276/pre-miR-276 ratio after *Heph* knockdown in *Drosophila* S2 cells (*n* = 6 biologically independent cell samples, each cell sample was from one well of a 48-well plate). Student's *t* test (two-tailed) was applied for two-group comparisons. The data are shown as the mean ± SEM. ** *P* < 0.01, *** *P* < 0.001. Source data and details of the statistical results are provided as a Source Data file.

miR-276 showed high homology among insects, particularly the "CU motifs" (Fig. 7a). We used S2 cells as a model system to study the nuclear export of *Drosophila* pre-miR-276. Knockdown of *Drosophila Heph*, the homologous gene of *Ptbp1* (Supplementary Fig. 4d), reduced the nucleocytoplasmic transport of pre-miR-276 (Fig. 7b) and the production of miR-276 in S2 cells (Fig. 7c). Therefore, the PTBP1-mediated nuclear export of pre-miR-276 may be conserved in insects.

## Discussion

The extensive mechanisms that impact parents' germline to facilitate the adaptation of offspring in response to high density are unclear. In the current study, high-population-density-induced expression of TF FOXN1 promotes expression of PTBP1 in oocytes that then enhances the expression of miR-276, which promotes synchronous hatching. By recognizing the "CU motifs" of pre-miR-276, PTBP1 recruits more XPO5 to facilitate nuclear export of pre-miR-276 in oocytes of gregarious locusts, thereby leading to hatching synchrony of progeny eggs (Fig. 8). Therefore, we propose a FOXN1–PTBP1 pathway that facilitates the nuclear export of pre-miRNAs in female oocytes in high density, to serve as a crucial basis for swarming and migration[16,50]. Furthermore,

the unreported PTBP1-promoted nuclear export of pre-miR-276 may be conserved in other insect species.

In our study, the response of female oocytes to crowding stimuli induces the hatching synchrony of progeny eggs. Previous research suggested that vibration may be the basic cue for egg-hatching synchrony, regardless of the population density[51]. However, our study reveals that the signal from female locusts is crucial for egg-hatching synchrony to adapt to high population density. Additionally, egg-hatching synchrony can prevent larval cannibalism[49], a phenomenon which is observed in hoppers (young nymphs) of migratory locusts[52]. Thus, egg-hatching synchrony may evolve as females whose offspring experience less cannibalism contribute more to the future gene pool. Furthermore, we report a pathway that high-density-promoted nuclear export of pre-miR-276 in the terminal oocytes mediates the transmission of parent experiences to offspring. Although miRNAs are considered as critical carriers of parental experiences of stress or chemical stimuli to alter the behavior or development of offspring[53,54], the extensive mechanisms of parental miRNA responses to environmental stimuli to regulate the offspring phenotype remain largely unclear. Given that the nuclear export of pre-miRNA is the key rate-limiting step for miRNA

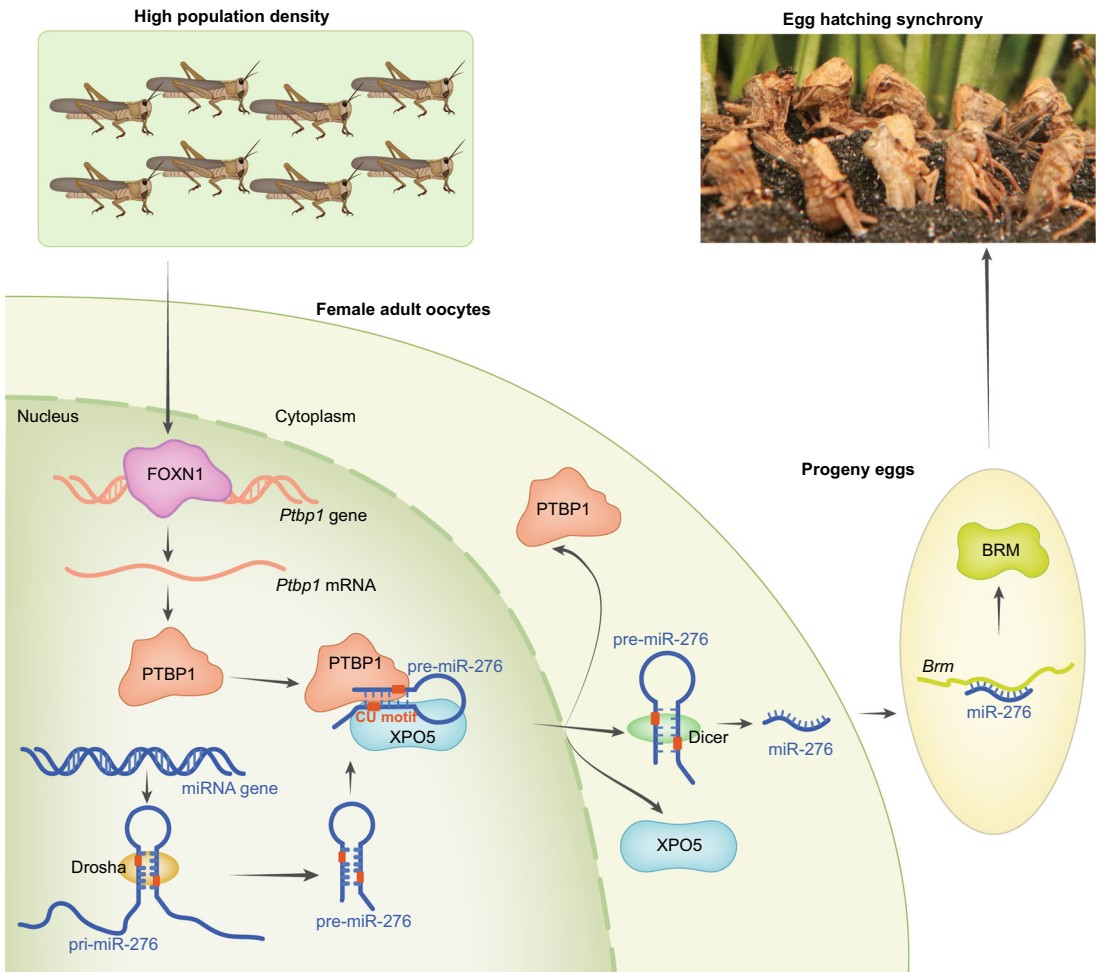

**Fig. 8 | Model of FOXN1-PTBP1–miR-276-regulated egg-hatching synchrony in response to high-density stimuli.** A transcriptional factor, FOXN1, is highly expressed in terminal oocytes of adult female gregarious locusts, and actives the transcription of polypyrimidine tract-binding protein-PTBP1, causing the high expression in terminal oocytes of adult female gregarious locusts. By recognition of "CU motifs" in pre-miR-276, PTBP1 recruits more XPO5 to pre-miR-276. Incorporation of PTBP1 and XPO5 promotes the nuclear transport of pre-miR-276, thereby increasing the production of mature miR-276. The activation of miR-276 and the downstream target BRM control the synchronous development of progeny eggs in gregarious locusts.

maturation[36], crowding stimuli are highly efficient in regulating miR-276 to mediate the transgenerational effects.

We uncovered a previously unreported role of FOXN1, a member of the forkhead (FOX) superfamily of TFs, in density-dependent transgenerational effects. FOXN1 responds to high density at the protein level rather than the mRNA level, similar to other TFs that mediate the signaling cascade of density-dependent phenotypic plasticity of locusts[41,55]. Post-transcriptional regulation facilitates the coordination of TF activity with external environmental changes in a precise and rapid manner[56,57]. A variety of regulatory factors may be involved in regulation of FOXN1 protein in response to density signals. One potential regulatory mechanism may be ubiquitination modification which can be induced in another FOX protein FOXO1 by insulin signal[58]. Other possible regulators could be miRNAs, which modulate the density-dependent phenotypic plasticity of locusts by post-transcriptional regulation of genes[8,16]. The underlying causes or possible upstream mechanisms linking environmental cues to FOXN1 protein warrant further investigation. Nevertheless, our study underscores the importance of TF in high density-induced egg-hatching synchrony by serving as upstream regulator of miR-276 in female terminal oocytes. Consistent with our findings, previous study showed that the networks of small RNA and TFs are crucial for environmentally-triggered transgenerational effects[59]. Although FOXN1 is well-known for its significance in the development and immunity of insects and mammals[60,61], a previous study suggested that the *Drosophila* homolog of FOXN1 JUMU is a maternal regulator for zygote genome activation[62]. Therefore, our study reveals a previously unknown role of FOXN1 in transmission of environmental signals from mother to offspring.

We discovered that an RNA binding protein PTBP1 is a previously unknown regulatory factor facilitating the pre-miRNA nucleo-cytoplasmic transport process. The XPO5-RanGTP complex is the indispensable exporter for most pre-miRNAs[36]; however, other regulatory factors besides XPO5 that participate in environment-regulated pre-miRNA nuclear export are largely undefined. Our findings of PTBP1-promoted nuclear export of pre-miRNA by coupling with XPO5 shed lights on the miRNA biogenesis. In addition, we reveal the unreported function of PTBP1 in reproductive strategies, as PTBP1 is well-known for its roles in neuronal development[63].

Moreover, the functions of PTBP1 exhibit differences yet share some similarities with XPO5 in miRNA biogenesis. On one hand, we found that distinct polypyrimidine tract-binding sites ("CU motifs") are essential for the recognition of pre-miRNAs by PTBP1. However, XPO5-mediated nuclear export is dependent on the stem–loop structure, which is a general characteristic of pre-miRNAs[64]. Changes in XPO5 caused by destructive stimuli affect the abundances of most miRNAs in animals[31,32,65]. In contrast, high density is not a destructive stimulus to locusts. Therefore, high density induces the expression of miRNAs in a

specific and precise manner, rather than through global changes of miRNA expression. In future studies, we will investigate the necessity of "CU motifs" in pre-miR-276 for egg-hatching synchrony, probably by developing lentiviral-based technology[66] to deliver pre- or pri-miRNAs into locusts. Additionally, we unexpectedly found that knockdown of *Ptbp1* resulted in the nuclear export of pre-miR-305 and pre-miR-34, two pre-miRNAs that lack "CU motifs", by increasing the binding of XPO5 with these two pre-miRNAs. We postulated that PTBP1 may repress other unidentified regulators capable of facilitating the nuclear export of these two pre-miRNAs. Thus, more studies are required to explore the regulatory mechanisms of nuclear export of these two pre-miRNAs. On the other hand, both PTBP1 and XPO5 are capable of engaging in various steps of miRNA biogenesis beyond nuclear export of pre-miRNAs. PTBP1 is involved in processing of pri- and pre-miRNAs in mammalian cells[67,68]. XPO5 plays crucial roles in miRNA processing steps and stability of pre-miRNAs[69], and is potentially involved in pre-miRNA processing by interacting with *Dicer*[70]. Thus, deletion of XPO5 can lead to entirely elimination of mature miRNAs[69], although the exportation efficiency of pre-miR-276 was just halved in our study. Therefore, further studies are required to explore the multiple roles of XPO5 and PTBP1 in miRNA biogenesis of locusts in response to density changes.

PTBP1-promoted nuclear export of pre-miR-276 is present in both locusts and *Drosophila*. Differing from in insects, PTBP1 prevents the processing of pri-/pre-miRNAs in mammalian cells, probably caused by the closeness of microprocessor binding sites and PTBP1 binding sites on pri-/pre-miRNAs[67,68]. The function discrepancy of PTBP1 between insects and mammals may be related to the differentiation of PTBP1 sequences or structures. However, further studies are needed to explore the differences of PTBP1-regulated miRNA biogenesis between insects and mammals.

The transduction pathways from parental density experiences to offspring hatching synchrony may involve multiple layers of regulatory mechanisms. We previously identified miR-276 targeting BRM in ovary as a crucial regulator of synchronous hatching of progeny eggs[16]. In the current study, the unreported FOXN1–PTBP1/XPO5 pathway serves as a key upstream regulatory mechanism for miR-276 in response to the external density signal. Overall, our findings, which range from upstream response mechanisms to downstream target genes, deepen the understanding of environment-induced transgenerational effects. In the future, more studies are required to delve into the intricate mechanisms of density-dependent transgenerational effects.

## Methods

### Insect husbandry
The locusts used in this study were from the same population at the Institute of Zoology, Chinese Academy of Sciences, Beijing, China. Well-ventilated cages (25 cm × 25 cm × 25 cm) were used to feed gregarious locusts at a density of ~300–400 insects per cage. Solitarious locusts were reared individually in separate and well-ventilated metal boxes with a volume of 10 cm × 10 cm × 25 cm. Both gregarious and solitarious locusts were reared under a photoperiod of 14 h/10 h light/dark at a temperature of 30 ± 2 °C. They were fed fresh greenhouse-grown wheat seedlings and wheat bran every day.

### Egg-hatching recording
Sexually mature locusts were mated in well-ventilated cages (25 × 25 × 25 cm) at densities of 40 pairs/cage and one pair/cage for gregarious and solitarious locusts, respectively. The eggs were produced by random selected multiple females during a 2-week oviposition period. The recording methods were performed according to previous study[16]. In detail, the egg pods were collected twice every day and incubated in sterilized sand with 10% pure water at 30 °C. The numbers of hatchlings were recorded four times every day.

### Quantitative PCR (qPCR) for mRNA and pri-, pre-, and mature miRNA
Terminal oocytes were dissected from sexually mature locusts with similar lengths (~4.5 mm), and the terminal oocytes were extruded using a forcep to avoid the contamination from the surrounding follicle cells. The terminal oocytes from one female locust were treated as one biological replicate. The eggs were collected within two hours after laying by female gregarious and solitarious locusts, and eggs from one egg pod were treated as one biological replicate. Total RNA was extracted by TRIzol reagent (Invitrogen). Partial pri-miR-276 sequences (~400 bp) that included pre-miR-276 based on the locust genome[35] was amplified, and was confirmed by gel image (Supplementary Fig. 15a) and sequencing. A miRcute miRNA First-Strand cDNA Synthesis Kit (Tiangen, China) and FastQuant First-Strand cDNA Synthesis Kit (Tiangen) were used to synthesize cDNA from pre-miRNAs or miRNAs and mRNAs or pri-miRNAs according to the manufacturers' instructions, respectively. qPCR was performed with a Roche LightCycler 480 by using a miRcute Plus miRNA qPCR Detection Kit (Tiangen) and a Real Master-Mix (SYBR Green) Kit (Tiangen). U6 snRNA and *Rp49* were used as endogenous controls for miRNAs or pre-miRNAs and mRNAs or pri-miRNAs, respectively. U6 snRNA and 18S rRNA of locusts were used as endogenous controls for nuclear and cytoplasmic pre-miRNAs, respectively. U27 snoRNA and 18S rRNA of *Drosophila* were used as endogenous controls of S2 cells for nuclear and cytoplasmic pre-miRNAs and miRNAs, respectively. U47 snoRNA and 18S rRNA of humans were used as endogenous controls in HEK 293T cells for nuclear and cytoplasmic pre-miRNAs and miRNAs, respectively. The amplification procedures followed the manufacturers' protocols, the melting curves were detected, and the PCR products were confirmed by sequencing. The cytoplasmic/nuclear expression ratio is calculated by dividing the relative expression in the cytoplasm by the relative expression in the nucleus. All primers used in this study are provided in Supplementary Data 1.

### Nuclear and cytoplasmic RNA extraction
Separation of the nuclear and cytoplasmic fractions of locust terminal oocytes, HEK 293T cells and S2 cells was performed according to previously described methods[16] with slight modifications, and all the following centrifugation steps were performed at 4 °C to avoid degradation. The tissues or cells were homogenized in cold PBS containing 0.2% Nonidet P-40, protease inhibitor (Thermo Fisher) and RNasin (Promega) and then centrifuged to remove the insoluble fragment at 200 × g for 10 min. The supernatant was centrifuged at 376 × g for 15 min to separate the nucleus and cytoplasm. Then, PBS containing 0.2% Nonidet P-40, protease inhibitor and RNasin was used to wash the nuclear pellet five times, and centrifugation was performed at 376 × g for 10 min. The cytoplasmic fraction in the supernatant was centrifuged at 376 × g for 15 min five times to remove the residual nuclei. Subsequently, RNA from the nuclear and cytoplasmic fractions was extracted with TRIzol reagent. U6 snRNA and 18S rRNA of locusts were used as nuclear marker and cytoplasmic marker in locust oocytes to detect the quality of the nuclear and cytoplasmic fractions, respectively. U27 snoRNA and 18S rRNA of *Drosophila* were used as nuclear and cytoplasmic markers in S2 cells to detect the quality of the nuclear and cytoplasmic fractions, respectively. U47 snoRNA and 18S rRNA of humans were used as nuclear and cytoplasmic markers in HEK 293T cells to detect the quality of the nuclear and cytoplasmic fractions, respectively. The original uncropped gel image and the unprocessed scans are shown in Supplementary Fig. 15b, d.

### In situ fluorescence hybridization
The oligo probes for pre-miR-276 hybridization were reverse-complemented with partial sequences of pre-miR-276 (~20 nt), containing the terminal loop but avoiding the mature miRNA sequences. The diagram for pre-miR-276 probe is shown in Supplementary Fig. 1.

The probes were modified with locked antisense nucleic acid (LNA) and labeled with digoxin (DIG) at both the 5′ and 3′ ends. The probes for pre-miR-67 of *C. elegans* with similar lengths and the same modifications as the probe of pre-miR-276 were used as negative control (NC). The oligos were synthesized by Shanghai Zhanbiao Biotechnology Co., Ltd. (China). The ovarioles were fixed in 4% (wt/vol) paraformaldehyde (PFA) at 4 °C overnight and then washed twice with PBS containing 0.3% Triton X-100 (PBST). Then, they were incubated in Enhanced Immunostaining Permeabilization Buffer (Beyotime, China) for 30 min to increase the permeability of ovarioles. Next, the ovarioles were digested with proteinase K (20 µg/ml) at 37 °C for 15 min, followed by hybridization with pre-miR-276 or NC probes (4 pmol/µl) at 37 °C overnight. Then, the ovarioles were successively washed in 2 × SSC, 1 × SSC, and 0.2 × SSC at 37 °C for 30 min before being incubated with a biotin-labeled anti-DIG antibody (BOSTER, China) at 4 °C overnight. Alexa Fluor™ 594-conjugated streptomycin (Invitrogen) (1:200) was applied to detect the signal of pre-miRNAs. Hoechst (1:2000, Thermo Fisher) was used to stain the nuclear DNA. Fluorescence images were captured with an LSM 710 confocal fluorescence microscope (Zeiss) at a magnification of 20×, and the intensity of the fluorescence signal was measured and analyzed with ImageJ software. The mean fluorescence intensities of all the oocytes from one female locust were treated as one biological replicate. All the sequences of RNA oligos are listed in Supplementary Data 2.

### RNA pull-down assays and mass spectrometry (MS)

RNA pull-down assays were performed under nuclease-free conditions. RNA oligos labeled with biotin at the 3′ end for pre-miR-276 (62 nt) and pre-miR-67 (69 nt) of *C. elegans* (negative control) were synthesized by RiboBio (Guangzhou, China). The protein samples from oocytes of three female locusts were treated as one biological replicate and were extracted with 600 µl of T-PER tissue protein extraction reagent (Thermo Fisher) with 1% (vol/vol) protease inhibitor (Thermo Fisher). Two biological replicates were performed. A 10-µl aliquot of the sample protein lysate was stored as "input". Streptavidin magnetic beads (50 µl) (Pierce™ Magnetic RNA-Protein Pull-Down Kit, Thermo Fisher) were prewashed with 0.1 M NaOH, 50 mM NaCl, and 100 mM NaCl in a 1.5 ml microcentrifuge tube and then washed three times using 20 mM Tris (pH 7.5). Subsequently, the beads were incubated with biotin-labeled pre-miR-276 or an NC probe (50 pmol) in 50 µl of 1 × RNA capture buffer at room temperature with agitation for 30 min. The beads conjugated with the probes were incubated with protein samples in a 100 µl RNA–protein binding reaction (10 × protein–RNA binding buffer, 50% glycerol and nuclear-free water) at 4 °C with rotation for one hour. After that, the beads were washed with 100 µl of 1 × wash buffer four times, and the proteins were eluted with 50 µl of elution buffer (40 mM NaCl, 10 mM Tris-HCl pH 7.2, 1 mM MgCl₂, and RNases A/T1) at 37 °C for 30 min. The proteins were separated by SDS–PAGE (12%) and detected by silver staining or analyzed by Western blotting. Two biological repeats and the original gel images of silver staining are shown in Supplementary Fig. 3.

To identify the proteins that bound with pre-miR-276, the bands that were specifically present in the pre-miR-276 group in PAGE were cut and subjected to MS (MS/MS) analysis at BPI (Beijing, China). Simultaneously, the corresponding parts of the gels for the NC group were subjected to MS/MS assay to control for nonspecific binding. The processed spectra were identified based on the locust protein database (http://159.226.67.243) through a MASCOT search. The MASCOT ion score for each peptide was calculated by [−10*log10($p$)], where $p$ is the probability that the observed match is a random event. The unique peptides with scores over the significance threshold level ($p < 0.05$) were used for protein identification. In this study, an individual ion score >20 was considered to indicate extensive homology. The candidate proteins were further analyzed with protein domain enrichment methods implemented in the LocustMine platform[41]. The identified proteins are listed in Supplementary Table 1. The raw data for mass spectrometry have been deposited at iProX Consortium (IPX0006309000/PXD041810). All the sequences of RNA oligos are listed in Supplementary Data 2.

### RNAi assay

RNAi experiments were performed by using double-stranded RNAs (dsRNAs) that were 300–500 bp in length and designed specifically for each gene. dsRNA of green fluorescent protein (ds*Gfp*) was used as the NC. All dsRNAs were synthesized by using the T7 RiboMAX Expression RNAi System (Promega) following the manufacturer's instructions. All injections were performed with a Nanoliter Injector 2000 (World Precision Instruments) at the dorsal site near the locust ovary, as previously described[16]. Female locusts were subjected to the first injection at the end of the fifth instar, and the subsequent injections were performed every five days. The dosage was 0.5 µg per locust each time. After three injections, some of the female locusts were sacrificed for sampling, and others were mated with male locusts. The mated females were continuously injected with dsRNAs every five days and were allowed to produce eggs freely.

The RNAi experiments in S2 cells were performed by dsRNA transfection of cells plated on 48-well plates. The dosage of dsRNAs was 2 µg per well, and the cells were transfected by using Lipofectamine 3000 (Invitrogen) in accordance with the manufacturer's instructions. The efficiency of RNAi is shown in Supplementary Fig. 4 and the primer sequences used for dsRNA synthesis are listed in Supplementary Data 1.

### Rescue experiments

Gregarious female locusts were injected with ds*Ptbp1* as described above. The next day, the locusts were injected with agomir-NC or agomir-276 according to the methods described above, and the agomir can be successfully delivered into locusts and can mimic the function of mature miR-276[16]. The subsequent injections, mating experiments and egg pod collection were performed in a manner similar to that described above.

### Western blot analysis

The antibody for locust PTBP1 was generated by immunizing a rabbit. The antigen of PTBP1 consisted of purified 6xHis-tagged antigenic fragments (Abclonal, China). The antibody for locust XPO5 was commercially supplied by Abclonal. The specificity of the antibody was confirmed by RNAi experiments in locusts (Supplementary Fig. 16). The total protein from the terminal oocytes and eggs, or nuclear–cytoplasmic protein of terminal oocytes was extracted with TRIzol reagent according to the manufacturer's instructions (Invitrogen). The proteins were separated by SDS–PAGE (10%) in SDS running buffer (EASYBIO) and then transferred to polyvinylidene difluoride (PVDF) membranes (Millipore) by using 1 × transfer buffer (Tanon). Then, the membranes were blocked by using 5% (wt/vol) skim milk at room temperature for 1 h. The membranes were incubated with primary antibodies [anti-PTBP1 antibody, 1:500, homemade; anti-BRM antibody, 1:500, homemade[16]; anti-FOXN1 antibody: 1:500, Proteintech; anti-Prd antibody, 1:500, Proteintech; anti-GAPDH antibody, 1:5000, homemade[10]; anti-H3 antibody, 1:5000, EASYBIO; anti-Tubulin antibody, 1:5000, EASYBIO; anti-XPO5 antibody (for HEK 293T cells, 1:500, Abcam; for locust, 1:500, Abclonal)] diluted in 5% skim milk at 4 °C overnight. Then, the membranes were incubated with secondary antibody (1:5000, EASYBIO) for one hour at room temperature. The immunological blot was detected with a SuperSignal West Femto Substrate Kit (Thermo Fisher). Band densitometry analysis was performed by using ImageJ software. The original uncropped blot image and the unprocessed scans were showed in Supplementary Figs. 17 and 18.

## Plasmid construction and cell transfection

The sequences of the *Ptbp1* gene and *Foxn1* gene (containing the whole ORF), partial pri-miR-276 (~400 bp), mutated pri-miRNA (~400 bp) and *Ptbp1*'s promoters (823 bp, 1391 bp, and 1866bp) were amplified by LA Taq DNA polymerase (TAKARA) by using primers designed according to the locust genome database[35]. The original images for amplification of promoter sequences are provided in Supplementary Fig. 15a. Subsequently, the sequences of *Ptbp1* gene, pri-miR-276 and mutated pri-miRNA were cloned into the pcDNA3.1(+) vector (Invitrogen) by using the BamHI and XhoI restriction sites. Sequences selected from the *LacZ* ORF were used as control of *Ptbp1*, and were constructed into the pcDNA3.1(+) vector by using KpnI and XhoI restriction sites. The sequence of *Foxn1* gene was cloned into the pAc5.1/V5-His B vector (Invitrogen) and the sequences of promoter were cloned into the pGL4.10 vector (Invitrogen) by using KpnI and XhoI. The amplification primers are listed in Supplementary Data 1.

HEK 293T cells plated on 48-well plates were transfected with the pri-miR-276 expression vector at a dose of 0.25 µg per well to determine the expression of mature miR-276. The pcDNA3.1(+) vector fused with *Ptbp1* or *LacZ* (0.2 µg) was co-transfected with the pri-miR-276 expression vector or empty pcDNA3.1(+) (NC) or mutant pri-miRNAs (0.1 µg). The mutation sites are shown in Fig. 5b. S2 cells plated on 48-well plates were co-transfected with pGL4.73 vector (200 ng) and pGL4.10 vector fused with putative promoter sequences of *Ptbp1* (200 ng). The empty plasmid of pGL4.10 and pGL4.73 vector were used as negative control for promoters, and internal control of transfection efficiency and luciferase activity, respectively. For verification of activation of FOXN1 on *Ptbp1*, S2 cells plate on 48-well plates were transfected with ds*Jumu* or ds*Gfp* (2 µg) for 24 h, and then the pGL4.73 vector fused with promoter of *Ptbp1* (sequence 2) was co-transfected with the FOXN1 expression vector or empty pAc5.1/V5-His B. All transfections were performed by using Lipofectamine 3000 in accordance with the manufacturer's instructions. The cells were subjected to total RNA extraction or nuclear/cytoplasmic RNA extraction or dual-luciferase report assay 48 h after transfection. For each treatment, five to six cell samples were collected and each cell sample was from one well of a 48-well plate.

## Amplification of sequences of transcription factors (TFs)

Detection of expression of TFs in different tissues was performed by amplification of TFs sequences according to locust genome[35], using the qPCR primer (Supplementary Data 1). The original images of amplification results are in Supplementary Fig. 15c.

## Dual-luciferase report assay

The dual-luciferase report assay was performed by using the Dual-Glo Luciferase Assay System (Promega) according to the manufacturer's instructions. The S2 cell transfection was performed as described above, forty-eight hours after co-transfection of promoter sequence and FOXN1, S2 cells were lysed by 5 × PLB lysis buffer at room temperature for 20 min. Subsequently, the lysate was mixed with Luciferase Assay Buffer II and Stop & Glo Buffer (Promega), and the luciferase activity was measured using a luminometer. The luciferase activity was calculated by the ratio of firefly luciferase activity (pGL4.10) to Renilla luciferase activity (pGL4.73).

## Electrophoretic mobility shift assay (EMSA)

EMSA was used to verify the binding of FOXN1 with *Ptbp1* promoter by using the LightShift Chemiluminescent EMSA Kit (Thermo Fisher). The double-stranded oligonucleotides labeled with biotin at the 5′-terminal were used as positive probes. Unlabeled probes (cold probes) and base mutation probes were used as competitive probes in Fig. 5l. Mutation probes with biotin at the 5′-terminal was used without wild probe in Supplementary Fig. 7. Nuclear proteins of the locust oocytes were extracted by using Nuclear and Cytoplasmic Protein Extraction Kit

(Beyotime). Nuclear protein extraction (20 µg) was incubated with 200 fmol biotin-labeled probes in binding buffer (2.5% glycerol, 5 mM MgCl$_2$, 1 M KCl, 50 ng poly (dI·dC), 0.05% NP-40, 200 mM EDTA) at room temperature for 20 min. For competition assay, the cold probes (400 pmol) or mutated probes (400 pmol) and anti-FOXN1 (Proteintech) antibody or anti-mouse IgG (Merckmillipore) were respectively added into the reaction mixtures. The protein-DNA complexes were separated using a 5% native polyacrylamide gel in cold 0.5 × TBE buffer and then transferred to nylon membrane (Thermo Fisher). The nylon membrane was then exposed to a UV light cross-linking instrument for 300 s (254 nm, 1200 mJ), and then was blocked for 15 min at room temperature. After that, the membrane was incubated with streptavidin-horseradish peroxidase conjugate for one hour at room temperature. After washing in 1 × washing buffer four times, the membrane was incubated with Substrate Equilibration Buffer for 5 min, and then the bands were detected by a SuperSignal West Femto Substrate Kit (Thermo Fisher). The sequences for all the probes were listed in Supplementary Data 2.

## Conduction of XPO5 knockout HEK 293T cell line

The XPO5-deleted HEK 293T cell line was conducted by using CRISPR–Cas9, and was performed by Ubigene Biosciences (Guangzhou, China). The sequences of gRNAs using for knockout of *Xpo5* are listed in Supplementary Data 1. The mutated nucleotides are shown in Supplementary Fig. 9, and the primers used for verification of the mutated sequences are listed in Supplementary Data 1.

## Co-IP assay

A co-IP assay was performed by using an Immunoprecipitation Assay Kit (Thermo Fisher) following the manufacturer's protocol. The magnetic beads were washed with Ab binding and washing buffer and then incubated with 5 µg of an anti-XPO5 antibody (ABclonal: for co-IP in terminal oocytes of locusts; Abcam: for co-IP in HEK 293T cells) or 5 µg of an anti-PTBP1 antibody with rotation for 30 min at room temperature. An anti-IgG antibody (rabbit or mouse) was used as an NC. For the co-IP experiment in locusts, terminal oocytes from one female locust were treated as one biological replicate. For the co-IP experiment in HEK 293T cells, cells (1 × 10$^7$) were collected 48 h after co-transfected with pri-miRNAs, NC, or mutated pre-miRNAs and PTBP1 or LacZ. Terminal oocytes or cells were homogenized in ice-cold RIPA lysis buffer with protease inhibitor. Protein samples were incubated with the bead–antibody complex for 2 h at 4 °C. Subsequently, the beads were washed with washing buffer three times, and the protein–bead complexes were eluted with elution buffer. The proteins were analyzed by Western blot assay as described above. The original uncropped gel image and the unprocessed scans were showed in Supplementary Fig. 17.

## RNA immunoprecipitation (RIP) assay

RIP experiments were performed using an RNA Immunoprecipitation Kit (BersinBio, China) following the manufacturer's protocol with slight modifications. Terminal oocytes from one locust were used as one biological replicate. Approximately 4 × 10$^7$ HEK 293T cells were used as one replicate. The tissues or cells were homogenized in polysome lysis buffer with protease inhibitor (Thermo Fisher) and RNase inhibitor (Promega), and then the lysates were stored at −80 °C for thorough cell lysis. The frozen protein homogenates were thawed quickly (~5 min) at room temperature. After centrifugation for 10 min at 4 °C (13,523 × $g$), 2% of the supernatant was used as input and stored at −80 °C. The remaining supernatant was divided for incubation with antibody of PTBP1 (5 µg), XPO5 (5 µg), or IgG (5 µg) at 4 °C overnight. The protein A/G beads (20 µl) were washed with polysome lysis buffer and then co-incubated with the protein–antibody complex for 6 h at 4 °C. Then, the complex was washed with polysome washing buffer 1 including DDT four times before being incubated with polysome

washing buffer 1 with DNase salt stock and DNase for 10 min at 37 °C. Three washes with polysome washing buffer 2 with DTT were performed at room temperature. After washing, the RNAs in the immunoprecipitates and input were extracted with TRIzol reagent (Invitrogen) and reverse-transcribed into cDNA using a miRcute First-Strand cDNA Synthesis Kit (Tiangen). qPCR was performed to determine the enrichment of pre-miRNAs in the PTBP1 or XPO5 complex by using a miRcute Plus miRNA qPCR Detection Kit (Tiangen). Ten percent of the lysate (input) and IgG controls were used for RNA quality assessment and standardization of the relative expression levels.

### Protein pull-down assay

The sequence of the *Ptbp1* gene (containing the whole ORF) was amplified with LA Taq DNA polymerase (TAKARA) and fused into the pET-28a vector by using the BamHI and XhoI restriction sites. The pET-28a vector without any insert gene was used as an NC. The *Escherichia coli* (*E. coli*) strain BL21 transformed with the plasmids was induced with isopropyl-β-D-thiogalactopyranoside (IPTG) (0.5 mM) at 16 °C overnight. After centrifugation, the cells were sonicated on ice. Subsequently, the lysate was separated into supernatant and sediment by centrifugation at 4 °C. The expression of PTBP1 in the supernatant was confirmed by Western blotting (Supplementary Fig. 8).

The supernatant was divided for incubation with protein A/G beads (40 µl), which were conjugated with an anti-PTBP1 antibody (5 µg) for 2 h at 4 °C. Then, the co-incubated proteins and beads were washed to remove nonspecific binding. After that, chemically synthesized pre-miR-276 (50 pmol) diluted with binding buffer (100 mM KCl, 20 mM Tris-HCl [pH 7.6], 0.1% Triton X-100, 0.1% Tween-20) was added. After incubation for 2 h at 4 °C, the RNAs were eluted with elution buffer containing 150 mM NaCl and 1% SDS for 15 min at room temperature. The RNA was extracted by TRIzol reagent (Invitrogen) and reverse-transcribed into cDNA using a miRcute First-Strand cDNA Synthesis Kit (Tiangen). qPCR was performed to determine the enrichment of pre-miR-276 by using the miRcute Plus miRNA qPCR Detection Kit (Tiangen).

### Immunostaining

For immunostaining in locusts, ovarioles dissected from female locusts were fixed in 4% (wt/vol) PFA at 4 °C overnight. For immunostaining in HEK 293T cells, the cells were collected 48 h after co-transfection of PTBP1 and pri-miR-276 or NC, and then fixed in 4% PFA for 5 min. Subsequently, the ovarioles or cells were washed with PBS containing 0.5% Triton X-100 (PBST) three times and then blocked with PBST containing 5% goat serum for one hour at room temperature. Then, the ovarioles or HEK 293T cells were incubated with primary antibody or IgG (negative control for PTBP1 in ovarioles) diluted in PBST containing 1% goat serum [anti-PTBP1 (1:500, homemade); XPO5 (1:500, Abcam); IgG (1:500, Merckmillipore)] at 4 °C overnight. After washing with PBST three times, the ovarioles or cells were incubated with Alexa Fluor 488 goat anti-rabbit IgG or Alexa Fluor 546 goat anti-mouse IgG (Invitrogen, 1:5000) for 30 min to detect the PTBP1 and XPO5 signals, respectively. Hoechst (Thermo Fisher, 1:2000) was used as the marker of nuclear DNA. The fluorescent images were captured by an LSM 710 confocal fluorescence microscope (Zeiss) at a magnification of 20× for ovarioles and 63× for HEK 293T cells. The ratio of co-localization of PTBP1 and XPO5 is calculated by dividing the number of co-localization cells by the number of PTBP1-positive cells.

The company names and catalog numbers for all commercial reagents in this study are in Supplementary Data 3.

### Bioinformatics analysis

Sequence alignment was performed with BioEdit software. The interactions of proteins were predicted with InterEvDock2[44–47]. The "CU motif" was identified as a sequence motif that was enriched in "CU" bases (≥5 tandem "C" or "U" bases). Two strategies were employed in predicting the transcription factors of *Ptbp1*. The first involved identifying TFs that can bind to the putative promoter region of *Ptbp1* (sequence 3: 1866 bp), using the MatInspector program[40] (http://www.genomatix.de/matinspector.html) with a cutoff of ≥0.95. The following selection of putative TFs was based on two criteria: 1) TFs that were present between −1391 bp to −823 bp, and 2) TFs that were not present between −1866 bp to −1391 bp. The second strategy involved analyzing the co-expression patterns of TFs with target genes using data from the LocustMine platform[41].

### Statistics and reproducibility

In two-group comparisons, Student's *t* test and Mann-Whitney *U* test were applied for analyzing the data with normal distribution and non-normal distribution, respectively. Kolmogorov-Smirnov test was used for normality test. All statistical results are in the "Source Data". One-way ANOVA followed by Tukey's post hoc tests with two-stage step-up method of Benjamini, Krieger and Yekutieli adjustment for multigroup comparisons. For analysis of hatching synchrony of entire eggs and hatching synchrony within each egg pod, one egg was treated as a biological replicate. The raw hatching times of eggs from each treatment were pooled, and the variation in egg-hatching time were compared by using Levene's test and F-test. For analysis of hatching synchrony within each egg pod, the raw hatching times from eggs of each treatment were standardized values in each egg clutch by subtracting the clutch mean. Subsequently, both the F-test and Levene's test were applied to analyze the hatching synchrony of egg pods from different treatments. One egg pod was regarded as a biological replicate in the analysis of the duration of hatching peaks. The data of hatching time in the frequency histogram of Supplementary Fig. 13a–13c and the hatching synchrony within egg pods in Supplementary Table 2 were sourced from our previous study[16] according to the copying license (https://www.pnas.org/author-center/publication-charges/standard-pnas-license-terms); previous data were re-analyzed and re-assembled in the present paper. All statistical analyses were performed with GraphPad Prism 8 software. Differences were considered statistically significant when *P* < 0.05.

The experiments in Figs. 1c, 3c, 4d, 4l, and Supplementary Fig. 8 were performed two or three times. The original images for these results are provided in the Supplementary Information file. Other experiments were performed one time with sufficient biologically independent samples.

### Reporting summary

Further information on research design is available in the Nature Portfolio Reporting Summary linked to this article.

## Data availability

The raw data for mass spectrometry have been deposited at iProX Consortium under accession code IPX0006309000/PXD041810. The published reference genome of migratory locust used for mapping is available at LocustBase (http://159.226.67.243). The published miRNA database is https://www.mirbase.org/. All the data needed to understand and assess the conclusions of this research are available in the article and Supplementary Information files. Source data are provided in this paper.

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

## Acknowledgements

We thank Dr. Xia Zhang and Dr. Pengcheng Yang for their help on cell experiments and bioinformatic analysis, respectively. We thank Dr. Xiaojiao Guo and Dr. Meiling Yang for their valuable advice on experimental design. This study was funded by the National Natural Science Foundation of China [32088102, 32270523], and the National Key R&D Program of China [2022YFD1400800 and 2022YFD1400500].

## Author contributions

Y.Z., J.H., W.G. and L.K. conceived and designed the experiments. Y.Z. performed the qPCR, Western blot, RIP, FISH, RNA pull-down, egg-hatching recording and cell experiments. J.H. and J.W. performed IP and immunostaining experiments. H.L. and Z.S. performed the prediction of protein interactions and provided the analysis tools. Y.Z., J.H., W.G. and L.K. analyzed the data. Y.Z., J.H., J.W. and L.K. wrote the manuscript.

## Competing interests

The authors declare no competing interests.
