## [Peer Review File · Nature Communications]

Parental experiences orchestrate locust egg hatching synchrony by regulating nuclear export of precursor miRNAREVIEWER COMMENTS

Reviewer #1 (Remarks to the Author):

Parental miRNAs, in response to environmental stimuli, function to regulate the phenotype of their offspring. Previous studies have shown that locust miR-276 promotes the synchronized hatching of progeny eggs, yet the mechanism linking environmental cues to miR-276 regulation within oocytes remains unclear. In the manuscript by Zhu et al., the authors propose that the FOXN1-PTBP1-XPO5 axis enhances the transport of pre-miR-276, thereby regulating the hatching synchrony of progeny eggs. Specifically, they suggest that PTBP1, up-regulated transcriptionally by FOXN1, expedites the translocation of pre-miR-276 from the nucleus to the cytoplasm, in conjunction with XPO5, in a 'CU motif'-dependent manner. Overall, these findings are potentially interesting. Most of the experiments are well designed, and many of the data presented are convincing. The subsequent comments are intended to offer suggestions for further improvement.

Major points:

1. The design of the EMSA assay in Fig. 3I doesn't convincingly demonstrate that the mutant probe has no competitive impact on the nuclear protein-binding capacity of the WT probe. To clarify, according to the Methods section, the mutant probe is also biotin-labeled, indicating that the signals in the gel images represent biotin intensity. Hence, regardless of whether the mutant probe can compete with the WT probe, an EMSA image will show a band of similar intensity to that of the WT probe group (lane 2).
2. The assertion in Fig. 5a that 'Ptpb1 knockdown did not reduce the nuclear export efficiencies of eight selected miRNA precursors' seems somewhat inaccurate. This is because the expression ratio (cytoplasmic/nuclear) of both pre-miR-305 and pre-miR-34 underwent significant changes when Ptpb1 was knocked down. It would be interesting to examine if there are any 'CU motifs' in pre-miR-305 and pre-miR-34 as well.
3. To determine the necessity of 'CU motifs' in pre-miR-267 (which are bound by PTBP1) for the reproductive strategy of locusts, it would be helpful if the rescue assays in Fig. 6 incorporated both mutant and WT pre-miR-267.

Minor points:

1. When the five candidate pre-miR-276-binding proteins are individually knocked down (as demonstrated in Fig. 2), it would be informative to display the relative expression levels of mature miR-276.
2. To effectively demonstrate that PTBP1 promotes the transport of pre-miR-276 in locust terminal oocytes, it would be beneficial to supplement the FISH imaging analysis in Fig. 2g with nuclear/cytoplasmic RNA extraction and RT-qPCR under the conditions of dsPtpb1 and control.
3. As seen in Fig. 3i-j, the protein level of FOXN1 in gregarious locust oocytes is three times higher than that in solitary locust oocytes, yet the mRNA level of Foxn1 is not affected by population density. The underlying causes or possible upstream mechanisms linking environmental cues to FOXN1 warrant further investigation.
4. Knockout of XPO5 halved the transport efficiency of pre-miR-276 (Fig. 4g) but entirely eliminated the expression of mature miR-276 (Fig. 4h). It appears that XPO5 might be involved in miRNA maturation beyond simply transporting pre-miRNA. The authors might want to consider omitting Fig. 4h to prevent potential misunderstanding, while discussing these data.
5. In Fig. 5, it could be valuable to conduct assays using a pre-miR-267 sequence with double mutation sites.

Reviewer #2 (Remarks to the Author):

Parental experiences orchestrate the hatching synchrony of offspring to promote aggregation

Overview: The goal of the paper is to illustrate a previously unknown pathway linking parental experience with offspring phenotypes in locusts. Specifically, the authors focus on the role of miRNAs in synchrony of offspring hatching in response to parental densities. Overall, I think that the experiments are well done, that the topic is interesting, and that the data are compelling. I also think that the authors could have done a better job of explaining the broader significance of their results.

Line 16: This is not always true – there are probably as many instances where parental effects negatively influence offspring as ones that benefit offspring.

Introduction: I didn't find that enough information was really provided here to allow the reader to understand the previous work done on miRNAs and their role in synchrony. For instance, what does it mean that "miR-276 upregulates the translation of BRM, resulting in synchronous development of progeny eggs"? It is probably explained in the paper that is referenced, but it is not obvious to the reader how individual levels of BRM lead to a group-wide phenotype.

Introduction: I felt that a better job could have been done in explaining what the new upstream steps of BRM regulation adds to our understanding of transgenerational effects. Characterizing these steps is amazing, for sure, but I didn't feel like the authors explained the greater significance of their findings to the broader community.

To be clear, I thought the authors did a very thorough job of testing each node of regulation of miRNA through their various experiments. I thought that using the RNA pull down assay for pre-mi-276 was a great approach, and systematically knocking down the resulting candidate genes in female locusts provided compelling evidence for Ptbp1 in mediating the mature/pre-miRNA ratio.

Line 145: This sentence is a bit misleading for readers: "Thus, Ptbp1 is involved in the maturation of miR-276 from pre-miR-276." It makes me think that the gene is directly involved in the maturation, but then the next paragraph goes on to explain the experiments demonstrating that the gene is involved in export (maturation occurs after it leaves the nucleus, but not through a direct interaction with Ptbp1, if I understand the results correctly). Just clarifying here that " Thus, Ptbp1 is involved in the maturation of miR-276 from pre-miR-276 through its influence on export." should help.

Line 181: "Thus, we knocked down the other three TFs one by one in terminal oocytes to verify their roles in Ptbp1 transcriptional activation." How did the authors knock down something specifically in the oocytes? Even just a basic statement about what stage they injected dsRNA, or where, would help, even if they don't go into the details here (I understand – it's the results section – but still, a little info is necessary).

Line 183: The authors say that knockdown of Prd doesn't decrease Ptbp1, but I see from the plot that it significantly (or near significantly) increases it. Does this mean that Prd might be a negative regulator of Ptbp1? What are the levels of Prd in solitary vs gregarious ovaries/oocytes?

Line 278: This was my favorite experiment in the whole study because it actually links the molecular mechanism to a phenotype (asynchrony).

Line 323: In this sentence and others, I think it would be better if the authors introduced the 'players' in the order of the pathway... such as "In in the current study, high-population density-induced expression of TF FOXN1 promotes expression of PTBP1 in oocytes that then enhances the expression of miR-276 which promotes synchronous hatching." Right now, the sentence reads as high density > PTBP1 > TF FOXN1 > miR-276 > gregarious locusts. This makes it hard for the reader to visualize the order of events, and also ends with a phenotype (gregarious locusts) that wasn't the phenotype quantified – more directly, the authors quantified hatching synchrony.

Discussion: The discussion ends rather abruptly; maybe it is a matter of style, but I feel like there should have been some summary statement to reiterate what was provided by the study when the all the results are taken into account.

Reviewer #3 (Remarks to the Author):

Environmental influences on development are ubiquitous, affecting many phenotypes in organisms. However molecular genetic and cellular mechanisms transducing environmental signals are still only barely understood, especially as they affect the development of animals. The manuscript by Zhu et al. describes experiments that reveal part of the intracellular mechanism in one of the most classic model systems of plasticity, the migratory locust. Therefore, this topic should be of broad general interest to biologists.

Overall, I think this is a detailed and comprehensive study. It would be asking too much for a complete mechanistic model, from environmental stimulus to phenotype, from one study. This manuscript significantly expands on a 2016 paper, also from the lab of Le Kang, that reported more proximate steps in the locust mechanism, and it also hints at broader evolutionary conservation. The experiments are sound, and the figures are well-designed. My most important critiques of this manuscript relates to its writing. The Results and Methods sections are well-written and can easily be followed. In fact, I'd like to praise the authors for the completeness of the Methods! However, the Introduction and Discussion are diminished somewhat by issues of word choice and sentence structures that can be unclear or unintentionally misleading.

Most importantly, the Intro lacks what I'd consider important contextualizing background. The paper would benefit from a few sentences on the broad importance of plasticity and highlighting the strengths of migratory locusts as a model. Some sections of the Discussion are redundant and focus a bit too much on summarizing the results. Instead, I'd suggest adding a paragraph to consider how the mechanistic model described in this study addresses the interesting, fundamental question of how variation (in this case, variation in time-to-hatching) can be regulated. This may help connect this study to the previous one by He et al. (2016) on the role of the transcription co-activator Brahma.

These seem like issues that can easily be addressed in revision. I've pointed to some of those problems by line number below.

- 1: The title of the manuscript does not really reflect its contents. That "parental experiences orchestrate the hatching synchrony of offspring to promote aggregation" has been known for *Locusta migratoria* for many years. Instead, I suggest a title that refers to the transduction mechanism of that plasticity or more directly summarizes that mechanism.
- 16: Delete "remarkably".
- 16-17: It would be useful to begin the Abstract making the point that plasticity is a general feature of developmental systems, not unique to locusts. -- Another framing the authors could use would be to pose the question of how timing synchrony is regulated.
- 18: As someone who does not work with locusts, I found myself wondering about the difference between eggs and egg pods and whether it's important here? Seems too granular for the Abstract.
- 20-21: Please rephrase. Using "high-density responsive" as an adjective for "transcription factor FOXN1" is awkward here since that fact is only established by the study, quite a ways into it.
- 26: Editing request: "...in order to foster..."
- 28: Editing request: "Overall, the discovery of the FOXN1-PTP1-miRNA pathway..."
- 33: Editing request: "The phenomenon..."
- 37-38: This sentence contains too many different ideas. Please split them up in order to clarify.
- 44-45: "...regulated by both coding- and non-coding RNAs." On their own, coding-sequence RNAs (mRNAs, right?) generally don't regulate anything, except by being translated into regulatory proteins. Please rephrase this statement to be more true to the basic biology.

- 49-50: There is a typo or grammatical error here that I can't identify in this sentence, but I'm not sure what point is being made.
- 62: Editing request: "...significant advances..."
- 99-100: "To confirm the identity of pri-miR-276 and pre-miR-276..." The experiment described in this paragraph doesn't seem to address the identity of these RNAs. Instead, I saw their value as confirming that RNAs of those sequences and structures (or in those reagents) can be properly processed into mature miRNA-276 in live animal cells.
- 104: It is not "at least 15 times higher". It is on average 15 times higher.
- 124 (and Fig 1i): Are the oocyte FISH images shown in Fig 1i from gregarious or solitary females?
- 162: "...so it is unable to directly respond to the changes of environmental signals." While I agree that's most likely, I think it's an untested assumption that should not be stated as fact.
- 183: Editing request: "...Prd and My do not affect..."
- 191: Editing request: "...effects of FOXN1 on transcription..."
- 193: Editing request: "...employing a biotin-labeled probe"
- 254-256: "Our results revealed that Ptbp1 knockdown did not reduce the nuclear export efficiencies of the eight selected miRNA precursors in the terminal oocytes (Fig. 5a)." But it looks like export of pre-miR-305 and pre-miR-34 was significantly increased. That's surprising! Obviously it's tangent to the main story in this report, but I'd be curious to know your interpretation of why that occurs. Perhaps as a short paragraph in the Supplemental Materials?
- 265: Editing request: "...affect the predicted stem-loop structure..."
- 282: It would be a kindness to readers to remind them here briefly of the previously identified role of Brahma in hatching synchrony.
- 290: Even as someone who works on insect development, "egg-hatching peak" was not something I was familiar with by that name. It may help to quickly define it. -- What does "(1090% hatching)" refer to?
- Fig 6a: The fitted curves are an effective way to communicate the change in distributions. However, in the interest of data transparency, I'd like to see the actual frequency histograms too.
- 297: The precise meaning of "egg pod" may not be clear to many readers. Importantly, why is a distinction being drawn between eggs and egg pods? Is this a potential factor creating non-independence in the data? It would help to explain a bit about the reproductive biology of locusts and to highlight the reason to focus on eggs or egg pods. It's also not clear to me whether the eggs in any particular analysis came from one or from multiple females, and whether that random factor was addressed in any way?
- 299 (and Fig 6e-i): The conclusion that there is greater variance in the hatching times of egg pods from solitary than from gregarious females seems very sound based on the SD's reported from multiple egg pods (is that the same as an egg clutch?). However, Fig 6e set off statistical alarm bells for me. I should say I'm a biologist, not a formally-trained statistician. But Student's t-test has a set of very limiting assumptions. It's not the best tool for a lot of the tests used in this study, but through most of the manuscript it didn't concern me too terribly. (Although you could consider, the Mann-Whitney U-test, Wilcoxon rank sum test or a two-sample permutation test as alternatives for most A-vs-B comparisons of mean measurements in this study.) However, SD is itself a summary statistic. Student's t test would be examining the difference in means of a group of statistics summarizing variance in separate groups (whose own distributions and means are not made available)? Instead, I suggest pooling the raw hatching times from eggs of each treatment and using the F-test to compare variances. And plot those actual hatching times. (This would require similar means for hatching times from different egg pods. If that's not the case, you could standardize values in each clutch by subtracting the clutch mean.)
- Fig 6f-i: What is the "polyphenic background" of the eggs used in these experiments? Were they from solitary or gregarious females? That point should be clarified.
- 312: Editing request: "...conserved mechanism in insects."
- Fig 7a: It would help readers to add a bit more to the species names. For most genomics work, abbreviations use one letter from the genus and 3 from the species, as in "Dmel" or "Lmig". But it would be an even greater kindness to readers to include the entire name.
- 328: Please provide a citation for the statement that synchronous hatching in the eggs of gregarious

females "increased fitness of progeny eggs." Keep in mind that for evolutionary biologists "fitness" has very specific meaning and should not be used casually. But I suspect there are classic papers describing differences in cannibalism among the morphs that can and should be cited in support of this statement.

- 332- 361: These two sections in the Discussion repeat several of the same points. I suspect they could be edited down.

- 338-339: The statement "Thus, response of parents to high-density signals is a crucial intrinsic regulatory mechanism for population homeostasis" suggests a sort of group selection adaptation that seems unwarranted. It seems more likely that this mechanism has evolved simply as females whose offspring experience less cannibalism contribute more to the future gene pool.

- 350: Editing request: "In the current study, we report a pathway..."

- 353: Editing request: "...chemical stimuli to alter the behavior..."

- 372-373: How many studies are being referenced here? There's only one citation at the end of the sentence.

- 374: Editing request: "Therefore, our study reveals a previously unknown role of FOXN1..."

- 389: Editing request: "In contrast, high density..."

- 392-393: "PTBP1-promoted nuclear export of pre-miR-276 is universally present in insects." The manuscript reports this interaction in two insect species. That cannot be called universal. (Especially if miR-276 is unique to insects!)

- Reporting Summary, Data section: I was unable to reach LocustBase at the URL provided.

Dave Angelini (ORCID 0000-0002-2776-2158)

REVIEWER COMMENTS

Reviewer #1 (Remarks to the Author):

Parental miRNAs, in response to environmental stimuli, function to regulate the phenotype of their offspring. Previous studies have shown that locust miR-276 promotes the synchronized hatching of progeny eggs, yet the mechanism linking environmental cues to miR-276 regulation within oocytes remains unclear. In the manuscript by Zhu et al., the authors propose that the FOXN1-PTBP1-XPO5 axis enhances the transport of pre-miR-276, thereby regulating the hatching synchrony of progeny eggs. Specifically, they suggest that PTBP1, up-regulated transcriptionally by FOXN1, expedites the translocation of pre-miR-276 from the nucleus to the cytoplasm, in conjunction with XPO5, in a 'CU motif'-dependent manner. Overall, these findings are potentially interesting. Most of the experiments are well designed, and many of the data presented are convincing. The subsequent comments are intended to offer suggestions for further improvement.

Major points:

1. The design of the EMSA assay in Fig. 3I doesn't convincingly demonstrate that the mutant probe has no competitive impact on the nuclear protein-binding capacity of the WT probe. To clarify, according to the Methods section, the mutant probe is also biotin-labeled, indicating that the signals in the gel images represent biotin intensity. Hence, regardless of whether the mutant probe can compete with the WT probe, an EMSA image will show a band of similar intensity to that of the WT probe group (lane 2).

We thank the Reviewer for this insightful comment. Indeed, the bind intensity in biotin-labeled mutant probe group cannot demonstrate the competitive impact on the nuclear protein-binding capacity. We now performed this experiment by using mutant probe without biotin, and found that the intensity of the band of the mutant group was similar to that of the wild type (WT)

probe group. Additionally, when employing biotin-labeled mutant probe without WT probe, the shift band was disappeared. We presented the new data in the revised manuscript (Figure 31 and Supplementary Fig. 7). We also revised the description for the probe in “Methods section” (lines 650-653).

2. The assertion in Fig. 5a that 'Ptbp1 knockdown did not reduce the nuclear export efficiencies of eight selected miRNA precursors' seems somewhat inaccurate. This is because the expression ratio (cytoplasmic/nuclear) of both pre-miR-305 and pre-miR-34 underwent significant changes when Ptbp1 was knocked down. It would be interesting to examine if there are any 'CU motifs' in pre-miR-305 and pre-miR-34 as well.

We thank the Reviewer for this comment. Based on the Method for analysis of “CU motifs” (lines 747-748), neither pre-miR-305 nor pre-miR-34 harbored “CU motifs” (Supplementary Fig. 12). Moreover, the two pre-miRNAs cannot bind with PTBP1, just like the other six pre-miRNAs we have chosen (Supplementary Fig. 10b). The elevated nuclear export efficiency of pre-miR-305 and pre-miR-34 following *Ptbp1* knockdown may be attributed to the increased binding of XPO5 to these two pre-miRNAs (Supplementary Fig. 10a). Hence, we speculated that PTBP1 may repress other unidentified regulators capable of facilitating the nuclear export of these two pre-miRNAs. However, further studies are required to explore the mechanisms of nuclear export of these two pre-miRNAs. We discussed this in the Discussion section of the revised main text (lines 427-432).

3. To determine the necessity of 'CU motifs' in pre-miR-267 (which are bound by PTBP1) for the reproductive strategy of locusts, it would be helpful if the rescue assays in Fig. 6 incorporated both mutant and WT pre-miR-267.

We thank the Reviewer for this comment. We performed rescue assays by overexpression of mature miR-276 because the agomir of miR-276 can be successfully delivered into locusts to mimic the function of miR-276¹. Indeed, we agree that the WT and mutant pre-miR-276 serve as the most suitable candidates for the rescue assays in Fig. 6 to determine the necessity of “CU motifs” in pre-miR-276 for egg-hatching synchrony. In order to introduce pre-miRNAs into locusts, we employed two approaches.

The first approach was injection of chemically synthesized pre-miRNAs (62 nt, with 2'-*O*-Me-modified at the 3' end) into locusts. The results showed that the production of mature miR-276 did not exhibit any augmentation following injection of pre-miR-276, although the abundance of pre-miR-276 was increased (Fig. 1a and 1b). Interestingly, this pre-miRNA can undergo processing into mature miR-276 in HEK 293T cells (Fig. 1c and 1d), ensuring that the sequence of pre-miR-276 we used was correct. Simultaneously, we confirmed the presence of exogenous pre-miR-276 in terminal oocytes of locusts and HEK 293T cells, by using pre-miR-276 with a biotin modification at the 5' end to prevent the potential interference from the mature miRNA (Fig. 1e and 1f). We also noticed that exogenous pre-miR-276 was unable to translocate into the nuclei of both terminal oocytes of locusts and HEK 293T cells (Fig. 1e and 1f), as we cannot specifically deliver pre-miRNAs into nuclei of cells. Therefore, injection of exogenous pre-miRNAs is inappropriate to studying the nuclear export of pre-miRNAs.

The second approach was injection of pri-miR-276 into locusts. The sequence of pri-miR-276 (~400 nt) used in this experiment was identical to that utilized in HEK 293T cells studies (Fig. 5), and transfection of vectors fused with pri-miR-276 can successfully produce mature miR-276 in HEK 293T cells (Fig. 1d). However, injection of pri-miR-276 neither led to an increase in the production of pre-miR-276 nor mature miR-276, despite the presence of increased pri-miR-276 levels in terminal oocytes of locusts (Fig. 1g-1i). Hence, exogenous pri-miR-276 also cannot be properly processed into pre-miR-276 and mature miR-276 in locusts.

In both of these approaches, we employed Entranster *in vivo* RNA transfection reagent (Engreen Biosystem Co., Ltd., China) as a vehicle for delivering pri- or pre-miRNAs into locusts, as we have previously achieved successful delivery of miRNAs and Piwi-interacting RNAs (piRNAs) into locusts using this liposomal transfection reagent^{2, 3}. The inability to production of mature miR-276 may be attributed to the unsuccessful releasing of pre- or pri-miRNAs from the liposomes, the incapacity of exogenous RNAs to translocate into nuclei of cells^{4, 5}, and other unidentified factors. Overall, unfortunately, we cannot utilize WT and mutant pre-miR-276 to determine the necessity of “CU motifs” for the reproductive strategy of locusts. Perhaps in future studies, we may develop lentiviral-based technology⁶ to deliver pre- or pri-miRNAs into locusts to address this issue. We now discussed this in Discussion section (lines 424-426).

Figure I. The processing of exogenous pri- or pre-miR-276 in locusts and HEK 293T cells. (a–b) Determine of the abundances of pre-miR-276 (a) and mature miR-276 (b) in terminal oocytes after 2d and 4d injection of pre-miR-276 in female locusts ($n = 5$, Student's t test, a: $P = 0.18$ for 2d, $P = 0.031$ for 4d; b: $P = 0.59$ for 2d, $P = 0.49$ for 4d). NC: negative control for pre-miR-276 (pre-miR-67 of *C. elegans* served as NC). (c–d) Determine of the abundances of pre-miR-276 (c) and mature miR-276 (d) in HEK 293T cells 12h, 24h, 36h and 48h after transfection of pre-miR-276 ($n = 5$, Student's t test, c: $P < 0.001$ for 12h, 24h, 36h and 48h; d: $P < 0.001$ for 12h, 24h, 36h and 48h). (e–f) Detection of exogenous pre-miRNAs in terminal oocytes of locusts and HEK 293T cells. NC: negative control for pre-miR-276 (pre-miR-67 of *C. elegans* without biotin modification). The nucleus of the terminal oocytes is delineated by a white dotted

line circle. Scale bar: 100 μm in e; 10 μm in f. Phalloidine and Hoechst were used to detect actin filaments and nucleus, respectively. (g–i) The expression of pri- (g), pre- (h), and mature miR-276 (i) in the terminal oocytes of locusts after 2d and 4d injection of pri-miR-276 in the female locusts ($n = 4$, Student's t test, g: $P = 0.05$ for 2d, $P = 0.020$ for 4d; h: $P = 0.64$ for 2d, $P = 0.31$ for 4d; i: $P = 0.52$ for 2d, $P = 0.64$ for 4d). The data are shown as the mean \pm SEM. * $P < 0.05$, *** $P < 0.001$.

Minor points:

1. When the five candidate pre-miR-276-binding proteins are individually knocked down (as demonstrated in Fig. 2), it would be informative to display the relative expression levels of mature miR-276.

We thank the Reviewer for this comment. We now provided the relative expression levels of mature miR-276 in Fig. 2e and the main text (lines 156-158). In fact, the relative expression level of mature miR-276 was decreased by 22% after knocking down *Ptbp1*, and was not changed after knocking down the other four genes. These results further supported the promotion effects of PTBP1 on the production of mature miR-276.

2. To effectively demonstrate that PTBP1 promotes the transport of pre-miR-276 in locust terminal oocytes, it would be beneficial to supplement the FISH imaging analysis in Fig. 2g with nuclear/cytoplasmic RNA extraction and RT-qPCR under the conditions of ds*Ptbp1* and control.

We thank the Reviewer for this comment. We have provided the RT-qPCR of pre-miR-276 in nuclear/cytoplasmic under the conditions of ds*Ptbp1* and control in the initial version of manuscript (Fig. 2f). Now we added the results of detection of nuclear/cytoplasmic marker (U6 and 18S) in Supplementary Fig. 5.

3. As seen in Fig. 3i-j, the protein level of FOXN1 in gregarious locust oocytes is three times higher than that in solitary locust oocytes, yet the mRNA level of Foxn1 is not affected by population density. The underlying causes or possible upstream mechanisms linking environmental cues to FOXN1 warrant further investigation.

We thank the Reviewer for this valuable comment. Indeed, the expression of FOXN1 were regulated by population density at protein levels in our study, similar to other TFs which mediate the signaling cascade of density-dependent phenotypic plasticity of locusts^{7, 8}. Post-transcriptional regulation facilitates the coordination of TF activity with external environmental changes in a precise and rapid manner^{8, 9}. A variety of regulatory factors may be involved in regulation of FOXN1 protein in response to density signals. One potential regulatory mechanism may be ubiquitination modification which can be induced in another FOX protein FOXO1 by insulin signal¹⁰. Other possible regulators could be miRNAs, which modulate the density-dependent phenotypic plasticity of locusts by post-transcriptional regulation of genes¹¹. The underlying causes or possible upstream mechanisms linking environmental cues to FOXN1 warrant further investigation. We now discussed this topic in the Discussion section (lines 390-400).

4. Knockout of XPO5 halved the transport efficiency of pre-miR-276 (Fig. 4g) but entirely eliminated the expression of mature miR-276 (Fig. 4h). It appears that XPO5 might be involved in miRNA maturation beyond simply transporting pre-miRNA. The authors might want to consider omitting Fig. 4h to prevent potential misunderstanding, while discussing these data.

We thank the Reviewer for this suggestion. Indeed, apart from transporting pre-miRNA, XPO5 is also involved in processing of pri-miRNA and may be required for the stability of pre-miRNAs¹². Additionally, the interaction between XPO5 with mRNA of Dicer, the enzyme responsible for cleaving pre-miRNAs into mature miRNAs¹³, implies the involvement of XPO5 in processing of pre-miRNAs. Hence, the diverse functions of XPO5 may lead to the elimination of mature miRNAs¹². According to the Reviewer's suggestion, we now omitted Fig. 4h and discussed this point in the Discussion section (lines 434-438).

5. In Fig. 5, it could be valuable to conduct assays using a pre-miR-267 sequence with double mutation sites.

We thank the Reviewer for this suggestion. Accordingly, we conducted the experiments

depicted in Fig. 5 by employing pre-miR-276 sequence with double mutation sites, including mutation of the both “CU motifs” (Mut 4), and mutation of one “CU motif” in conjunction with the “non-CU motif” (Mut 5 and Mut 6). All pre-miRNAs generated from these double mutation vectors were unable to bind with PTBP1 (Fig. 5j). Moreover, all these double mutations impeded the promotion effects of PTBP1 on the nuclear export of pre-miRNAs (Fig. 5f), and hindered the enhancement effects of PTBP1 on the binding affinity between XPO5 and pre-miRNAs (Fig. 5l). These results were consistent with the data obtained from single mutation assays, as the mutation of any one of the “CU motifs” can block the binding of PTBP1 to pre-miRNAs and impair the promotion effects of PTBP1 on nuclear export of pre-miRNAs. One unexpected result was that double mutation of the two “CU motifs” (Mut 4) or the “CU motif” on the 5p strand in conjunction with the “non-CU motif” (Mut 5) failed to yield mature miRNAs (Supplementary Fig. 11), probably due to the disruptions in certain biogenesis processes. Nevertheless, all the vectors with double mutations can produce pre-miRNAs (Fig. 5d), enabling us to detect the nuclear export of pre-miRNAs. Overall, the results of double mutations did not change our previous conclusions. Thus, both “CU motifs” in 5p and 3p strands of pre-miR-276 are required for PTBP1-mediated nuclear export. We now incorporated the data of double mutation assays into Fig. 5 and the main text (lines 289-302).

Reviewer #2 (Remarks to the Author):

Parental experiences orchestrate the hatching synchrony of offspring to promote aggregation

Overview: The goal of the paper is to illustrate a previously unknown pathway linking parental experience with offspring phenotypes in locusts. Specifically, the authors focus on the role of miRNAs in synchrony of offspring hatching in response to parental densities. Overall, I think that the experiments are well done, that the topic is interesting, and that the data are compelling. I also think that the authors could have done a better job of explaining the broader significance of their results.

Line 16: This is not always true – there are probably as many instances where parental effects negatively influence offspring as ones that benefit offspring.

We thank the Reviewer for this comment. Indeed, parental effects negatively influence offspring in many instances. We now revised the sentences to “Parental experiences generally affect the phenotypic plasticity of offspring” (line 17).

Introduction: I didn’t find that enough information was really provided here to allow the reader to understand the previous work done on miRNAs and their role in synchrony. For instance, what does it mean that “miR-276 upregulates the translation of BRM, resulting in synchronous development of progeny eggs”? It is probably explained in the paper that is referenced, but it is not obvious to the reader how individual levels of BRM lead to a group-wide phenotype.

We apologize for our unclear description of the previous study about the miR-276 regulated egg-hatching synchrony. The high expression of miR-276 in ovaries of female locusts, controls the egg-hatching synchrony of progeny eggs by upregulating a co-transcriptional factor BRM. By recognition of the stem-loop structure of the precursor mRNA of *Brm*, miR-276 enhances the translation of BRM although does not change the mRNA expression level¹. BRM is essential for the early development of embryos in *Drosophila*¹⁴, and is critical for nucleosome organization and subsequent transcription homeostasis¹⁵. Elevated expression of BRM in individuals lead to a high level of BRM in the group of gregarious locusts. Thus, high expression of BRM in locust group contributes to the developmental homeostasis of embryos, ultimately minimizing the variation in egg-hatching times of gregarious locusts. Now we introduced the roles of miR-276 and BRM in egg-hatching synchrony (lines 75-82).

Introduction: I felt that a better job could have been done in explaining what the new upstream steps of BRM regulation adds to our understanding of transgenerational effects. Characterizing these steps is amazing, for sure, but I didn’t feel like the authors explained the greater significance of their findings to the broader community.

We thank the Reviewer for this valuable comment. We added the explanations of the significance of our findings about new upstream steps of miR-276 and BRM (lines 63-68 and

lines 82-86).

To be clear, I thought the authors did a very thorough job of testing each node of regulation of miRNA through their various experiments. I thought that using the RNA pull down assay for pre-mi-276 was a great approach, and systematically knocking down the resulting candidate genes in female locusts provided compelling evidence for *Ptbp1* in mediating the mature/pre-miRNA ratio.

Thanks for the positive evaluation.

Line 145: This sentence is a bit misleading for readers: “Thus, *Ptbp1* is involved in the maturation of miR-276 from pre-miR-276.” It makes me think that the gene is directly involved in the maturation, but then the next paragraph goes on to explain the experiments demonstrating that the gene is involved in export (maturation occurs after it leaves the nucleus, but not through a direct interaction with *Ptbp1*, if I understand the results correctly). Just clarifying here that “Thus, *Ptbp1* is involved in the maturation of miR-276 from pre-miR-276 through its influence on export.” should help.

We thank the Reviewer for this comment. We now revised the sentence according to the advice (lines 158-159).

Line 181: “Thus, we knocked down the other three TFs one by one in terminal oocytes to verify their roles in *Ptbp1* transcriptional activation.” How did the authors knock down something specifically in the oocytes? Even just a basic statement about what stage they injected dsRNA, or where, would help, even if they don’t go into the details here (I understand – it’s the results section – but still, a little info is necessary).

We thank the Reviewer for this comment. Indeed, we cannot specifically knockdown genes in terminal oocytes of locusts. In fact, we injected the dsRNAs in at the dorsal site near the locust ovary by using Nanoliter Injector, in order to deliver the dsRNAs locally to the ovary as possible as we can. We detected the expression levels of the TFs and *Ptbp1* in terminal oocytes after three times of injections. We now added a basic statement about the injection stage and injection site in the results (lines 197-200).

Line 183: The authors say that knockdown of Prd doesn't decrease Ptbp1, but I see from the plot that it significantly (or near significantly) increases it. Does this mean that Prd might be a negative regulator of Ptbp1? What are the levels of Prd in solitary vs gregarious ovaries/oocytes?

We thank the Reviewer for this comment. According to the advice of the reviewer, we determined the expression of Prd in oocytes of gregarious and solitary locusts. The results showed that both the mRNA and protein levels of Prd in terminal oocytes were not significantly different between gregarious and solitary locusts. These results indicated that Prd may not be a regulator of *Ptbp1*. We added these data in the Results to exclude the regulation of Prd on *Ptbp1* (Supplementary Fig. 6; main text lines 201-202 and lines 204-207). The reason for the increase of *Ptbp1* by knockdown of Prd is probably related to the variation of the relative expression level of *Ptbp1* in *dsPrd* group.

Line 278: This was my favorite experiment in the whole study because it actually links the molecular mechanism to a phenotype (asynchrony).

Thanks for the positive evaluation.

Line 323: In this sentence and others, I think it would be better if the authors introduced the 'players' in the order of the pathway... such as "In in the current study, high-population density-induced expression of TF FOXN1 promotes expression of PTBP1 in oocytes that then enhances the expression of miR-276 which promotes synchronous hatching." Right now, the sentence reads as high density > PTBP1 > TF FOXN1 > miR-276 > gregarious locusts. This makes it hard for the reader to visualize the order of events, and also ends with a phenotype (gregarious locusts) that wasn't the phenotype quantified – more directly, the authors quantified hatching synchrony.

We thank the Reviewer for this suggestion. We now revised this sentence according to the advice (lines 365-367).

Discussion: The discussion ends rather abruptly; maybe it is a matter of style, but I feel like there should have been some summary statement to reiterate what was provided by

the study when the all the results are taken into account.

We thank the Reviewer for this valuable suggestion. We added a paragraph to summary the findings and addressed the significance of the study in the end of Discussion section (lines 447-455).

Reviewer #3 (Remarks to the Author):

Environmental influences on development are ubiquitous, affecting many phenotypes in organisms. However molecular genetic and cellular mechanisms transducing environmental signals are still only barely understood, especially as they affect the development of animals. The manuscript by Zhu et al. describes experiments that reveal part of the intracellular mechanism in one of the most classic model systems of plasticity, the migratory locust. Therefore, this topic should be of broad general interest to biologists.

Overall, I think this is a detailed and comprehensive study. It would be asking too much for a complete mechanistic model, from environmental stimulus to phenotype, from one study. This manuscript significantly expands on a 2016 paper, also from the lab of Le Kang, that reported more proximate steps in the locust mechanism, and it also hints at broader evolutionary conservation. The experiments are sound, and the figures are well-designed. My most important critiques of this manuscript relates to its writing. The Results and Methods sections are well-written and can easily be followed. In fact, I'd like to praise the authors for the completeness of the Methods! However, the Introduction and Discussion are diminished somewhat by issues of word choice and sentence structures that can be unclear or unintentionally misleading.

Most importantly, the Intro lacks what I'd consider important contextualizing background. The paper would benefit from a few sentences on the broad importance of plasticity and highlighting the strengths of migratory locusts as a model. Some sections of the Discussion are redundant and focus a bit too much on summarizing the results. Instead, I'd suggest adding a paragraph to consider how the mechanistic model described

in this study addresses the interesting, fundamental question of how variation (in this case, variation in time-to-hatching) can be regulated. This may help connect this study to the previous one by He et al. (2016) on the role of the transcription co-activator Brahma.

These seem like issues that can easily be addressed in revision. I've pointed to some of those problems by line number below.

Thanks for the positive evaluation and the valuable suggestions. We revised the Introduction section to describe the previous study¹ to help the readers to understand the function of miR-276 and BRM in egg-hatching synchrony (lines 63-68). We now highlighted the broad importance of plasticity and transgenerational effects, and highlighted the strengths of migratory locusts as a model (lines 35-40 and lines 82-86). We also revised the Discussion to make this part be more concise, according to the comments of this reviewer (lines 374-388). Simultaneously, we added a paragraph to address the significance of our findings about new upstream steps of miR-276 and BRM (lines 447-455).

- 1: The title of the manuscript does not really reflect its contents. That "parental experiences orchestrate the hatching synchrony of offspring to promote aggregation" has been known for *Locusta migratoria* for many years. Instead, I suggest a title that refers to the transduction mechanism of that plasticity or more directly summarizes that mechanism.

We thank the Reviewer for this comment. We now revise the title to "Parental experiences orchestrate the hatching synchrony of offspring by regulating nuclear export of precursor miRNA" to refer to the transduction mechanism.

- 16: Delete "remarkably".

We thank the Reviewer for this comment. We now deleted "remarkably" (line 17).

- 16-17: It would be useful to begin the Abstract making the point that plasticity is a general feature of developmental systems, not unique to locusts. -- Another framing the authors could use would be to pose the question of how timing synchrony is regulated.

We thank the Reviewer for valuable suggestion. We now incorporated the general importance of plasticity into the first sentence of the Abstract (line 17), due to the word limitation of Abstract. The significance of plasticity and transgenerational effects of locusts is highlighted in the second sentence of Abstract (lines 17-19).

- 18: As someone who does not work with locusts, I found myself wondering about the difference between eggs and egg pods and whether it's important here? Seems too granular for the Abstract.

Sorry for the confusion of the description. Locusts lay eggs in clusters known as egg pods¹⁶, which are synonymous with egg clutches. Here the “egg-pod number” means the number of egg pods (egg clutch), and “egg-number” indicates the number of eggs per pod. In our previous study, both of the two characteristics are affected by the density of parental locusts^{2, 3}. In order to avoid any potential confusion, we now deleted “egg pods” here (lines 18).

- 20-21: Please rephrase. Using "high-density responsive" as an adjective for "transcription factor FOXN1" is awkward here since that fact is only established by the study, quite a ways into it.

We thank the Reviewer for this comment. We revised this sentence to “we find that transcription factor Forkhead box protein N1 (FOXN1) responds to high population density and activates the *polypyrimidine tract-binding protein (Ptbp1)* in locusts” (lines 20-22).

- 26: Editing request: "...in order to foster..."

We thank the Reviewer for this comment. We revised the sentence according to this suggestion (line 26).

- 28: Editing request: "Overall, the discovery of the FOXN1-PTP1-miRNA pathway..."

We thank the Reviewer for this comment. We now revised this sentence and integrated this sentence with the previous one, owing to the word limitation of Abstract (lines 27-29).

- 33: Editing request: "The phenomenon..."

We thank the Reviewer for this comment. We revised the sentence according to this suggestion (line 31).

- 37-38: This sentence contains too many different ideas. Please split them up in order to clarify.

We thank the Reviewer for this comment. We revised the sentence according to this suggestion (lines 35-36).

- 44-45: "...regulated by both coding- and non-coding RNAs." On their own, coding-sequence RNAs (mRNAs, right?) generally don't regulate anything, except by being translated into regulatory proteins. Please rephrase this statement to be more true to the basic biology.

We thank the Reviewer for this comment. We revised this statement to "regulated by both proteins and non-coding RNAs" (lines 42-43).

- 49-50: There is a typo or grammatical error here that I can't identify in this sentence, but I'm not sure what point is being made.

We apologize for the errors in this sentence. We now revised to "Focused evaluation of such transgenerational effects may reveal novel mechanism of inheritance" (lines 47-48).

- 62: Editing request: "...significant advances..."

We thank the Reviewer for this comment. We revised the sentence according to this suggestion (line 60).

- 99-100: "To confirm the identity of pri-miR-276 and pre-miR-276..." The experiment described in this paragraph doesn't seem to address the identity of these RNAs. Instead, I saw their value as confirming that RNAs of those sequences and structures (or in those reagents) can be properly processed into mature miRNA-276 in live animal cells.

We thank the Reviewer for this comment. We revised the sentence according to this suggestion (lines 109-111).

- 104: It is not "at least 15 times higher". It is on average 15 times higher.

We thank the Reviewer for this comment. We revised this statement according to this suggestion (line 115).

- 124 (and Fig 1i): Are the oocyte FISH images shown in Fig 1i from gregarious or solitary females?

The oocyte FISH images shown in previous Fig 1i was from gregarious females. We now added another FISH images from solitary females (Fig 1i). Additionally, we removed the images of negative control (NC) into Supplementary Fig. 2.

- 162: "...so it is unable to directly respond to the changes of environmental signals." While I agree that's most likely, I think it's an untested assumption that should not be stated as fact.

We thank the Reviewer for this comment. We now revised this statement to "As an RNA binding protein, PTBP1 typically serves as a downstream effector of signal transduction, leading us to speculate that other upstream regulators might act on PTBP1 in response to high density" (lines 175-177).

- 183: Editing request: "...Prd and My do not affect..."

We thank the Reviewer for this comment. We revised the sentence according to this suggestion, and pointed out the slight increase of *Ptbp1* by knockdown of *Prd* according to the comment of Reviewer #2 (lines 200-202).

- 191: Editing request: "...effects of FOXN1 on transcription..."

We thank the Reviewer for this comment. We revised the sentence according to this suggestion (line 212).

- 193: Editing request: "...employing a biotin-labeled probe"

We thank the Reviewer for this comment. We revised the sentence according to this suggestion (line 214).

- 254-256: "Our results revealed that Ptbp1 knockdown did not reduce the nuclear export efficiencies of the eight selected miRNA precursors in the terminal oocytes (Fig. 5a)." But it looks like export of pre-miR-305 and pre-miR-34 was significantly increased. That's surprising! Obviously it's tangent to the main story in this report, but I'd be curious to know your interpretation of why that occurs. Perhaps as a short paragraph in the Supplemental Materials?

We thank the Reviewer for this comment. The sequences of pre-miR-305 and pre-miR-34 do not harbor "CU motif", as we respond to Reviewer #1 (Comment #2 in Major points). Based on the Method for analysis of "CU motifs" (lines 747-748), neither pre-miR-305 nor pre-miR-34 harbored "CU motifs" (Supplementary Fig. 12). Moreover, the two pre-miRNAs cannot bind with PTBP1, just like the other six pre-miRNAs we have chosen (Supplementary Fig. 10b). The elevated nuclear export efficiency of pre-miR-305 and pre-miR-34 following *Ptbp1* knockdown may be attributed to the increased binding of XPO5 to these two pre-miRNAs (Supplementary Fig. 10a). Hence, we speculated that PTBP1 may repress other unidentified regulators capable of facilitating the nuclear export of these two pre-miRNAs. However, further studies are required to explore the mechanisms. We discussed this in the revised main text (lines 427-432).

- 265: Editing request: "...affect the predicted stem-loop structure..."

We thank the Reviewer for this comment. We revised the sentence according to this suggestion (line 292).

- 282: It would be a kindness to readers to remind them here briefly of the previously identified role of Brahma in hatching synchrony.

We thank the Reviewer for this comment. We added the function of Brahma (BRM) in hatching synchrony here (lines 313-316). In addition, we added a detailed introduction about the roles of BRM in hatching synchrony (lines 75-82), as we respond to Reviewer #2.

- 290: Even as someone who works on insect development, "egg-hatching peak" was not something I was familiar with by that name. It may help to quickly define it. -- What does "(1090% hatching)" refer to?

We apologize for the mistake in this sentence. The “egg-hatching peak” indicates the 10–90% egg hatching, as we previously used this parameter¹. We omitted a “–” in previous version of manuscript and now we revised it (line 318). We used the time duration of egg-hatching peak to represent the degree of hatching synchrony within each egg pods, to support our findings regarding the fitted curves presented in Figures 6a and 6c.

- Fig 6a: The fitted curves are an effective way to communicate the change in distributions. However, in the interest of data transparency, I'd like to see the actual frequency histograms too.

We thank the Reviewer for this comment. We now provided actual frequency histograms in the Supplementary Fig. 13.

- 297: The precise meaning of "egg pod" may not be clear to many readers. Importantly, why is a distinction being drawn between eggs and egg pods? Is this a potential factor creating non-independence in the data? It would help to explain a bit about the reproductive biology of locusts and to highlight the reason to focus on eggs or egg pods. It's also not clear to me whether the eggs in any particular analysis came from one or from multiple females, and whether that random factor was addressed in any way?

We thank the Reviewer for this comment. Sorry for the unclear description of “eggs and egg pods”. Locusts lay eggs in clusters known as egg pods¹⁶, which are synonymous with egg clutches. We now explained this in the main text (line 339). We employed the term “egg-hatching synchrony” to denote the entire hatching synchrony of eggs, and “hatching synchrony within each egg pod” to indicate the degree of synchrony in hatching time within each egg pod (egg clutch). The hatching synchrony within egg pods is a crucial factor in ensuring the overall synchrony of the eggs. Moreover, considering the significance of hatching synchrony within egg pods for avoiding cannibalism¹⁷, so we analyzed the hatching synchrony within egg pods.

We now explained the reason why we focus on hatching synchrony within egg pods (lines 340-342). However, to prevent any confusion between the egg-hatching synchrony and hatching synchrony within each egg pod, we removed the results of the latter to Supplementary Materials (Supplementary Table 2). In fact, the eggs used for analyzing the two parameters were produced by random selected multiple females during a two-week oviposition period (Methods section, lines 468-469). To analyze “egg-hatching synchrony”, we pooled the raw hatching times of eggs from each treatment, and compared the variation in egg-hatching time by using Levene’s test and F-test (Figure legends in Supplementary Fig. 13). One egg was treated as a biological replicate for the normal curve fitting of hatching time, Levene’s test and F-test. Differently, for “hatching synchrony within each egg pod”, we compared the variation of hatching time within each egg pods. One egg pod was treated as a biological replicate. Now we deleted the Student’s *t*-test of SD according to the next comment of this reviewer. All the analysis methods were described in detail in the Methods section (lines 760-772).

- 299 (and Fig 6e-i): The conclusion that there is greater variance in the hatching times of egg pods from solitarious than from gregarious females seems very sound based on the SD's reported from multiple egg pods (is that the same as an egg clutch?). However, Fig 6e set off statistical alarm bells for me. I should say I'm a biologist, not a formally-trained statistician. But Student's t-test has a set of very limiting assumptions. It's not the best tool for a lot of the tests used in this study, but through most of the manuscript it didn't concern me too terribly. (Although you could considered, the Mann-Whitney U-test, Wilcoxon rank sum test or a two-sample permutation test as alternatives for most A-vs-B comparisons of mean measurements in this study.) However, SD is itself a summary statistic. Student's t test would be examining the difference in means of a group of statistics summarizing variance in separate groups (whose own distributions and means are not made available)? Instead, I suggest pooling the raw hatching times from eggs of each treatment and using the F-test to compare variances. And plot those actual hatching times. (This would require similar means for hatching times from different egg pods. If that's not the case, you could standardize values in each clutch by subtracting the clutch mean.)

We thank the Reviewer for this comment. We agree that Student's *t* test is inappropriate to compare the SD, which is a summary statistic. We now revised this analysis according to the Reviewer's suggestion (including the previous Fig. 6e–6i). In specific, we pooled the raw hatching times from eggs of each treatment and standardized values in each egg clutch by subtracting the clutch mean. Subsequently, we utilized both the F-test and Levene's test to analyze the hatching synchrony of egg pods from different treatments. The outcomes obtained from the two statistical methods did not change our previous conclusions. By the way, we plotted all the actual egg-hatching times by using frequency histograms (Supplementary Fig. 13), as we respond to the previous comment related to Fig. 6a. To prevent any confusion between the egg-hatching synchrony and hatching synchrony within each egg pod, we removed the results of the latter to Supplementary Materials (Supplementary Table 2). Additionally, we checked all of the two-group comparisons in this study, and we revised some of the statistical methods. Student's *t* test and Mann-Whitney *U* test were applied for analyzing the data with normal distribution and non-normal distribution, respectively. The revisions did not change our previous conclusions. The statistical methods for two-group comparisons were mentioned in the Figure legends, and all the analysis methods were described in detail in the Methods section (lines 756-773).

- Fig 6f-i: What is the "polyphenic background" of the eggs used in these experiments? Were they from solitary or gregarious females? That point should be clarified.

The eggs in previous Fig. 6f, 6h, and 6i were from the gregarious females, who displayed higher expression levels of miR-276 and *Ptbp1*. The eggs in Fig. 6g were from the solitary females, who displayed lower expression levels of miR-276. We described the "polyphenic background" of the eggs in the results of previous version of manuscript (Previous manuscript, page 11, lines 299–305). Notably, the fitted curves and the results of Levene's test in Fig. 6f–6i have been presented in our previous publication¹, we now re-analyzed these data according to the previous comment of this reviewer (Supplementary Table 2), and plotted the actual egg-hatching times by using frequency histograms (Supplementary Fig. 13).

- 312: Editing request: "...conserved mechanism in insects."

We thank the Reviewer for this comment. We revised the sentence according to this suggestion (line 355).

- Fig 7a: It would help readers to add a bit more to the species names. For most genomics work, abbreviations use one letter from the genus and 3 from the species, as in "Dmel" or "Lmig". But it would be a even greater kindness to readers to include the entire name.

We thank the Reviewer for this comment. We revised the abbreviations according to this suggestion (Fig. 7a), and we provided the entire names in the Figure legends of Fig. 7a.

- 328: Please provide a citation for the statement that synchronous hatching in the eggs of gregarious females "increased fitness of progeny eggs." Keep in mind that for evolutionary biologists "fitness" has very specific meaning and should not be used casually. But I suspect there are classic papers describing differences in cannibalism among the morphs that can and should cited in support of this statement.

We thank the Reviewer for this valuable comment. We now deleted "fitness" here, and revised this sentence to "we propose a FOXN1–PTBP1 pathway that facilitates the nuclear export of pre-miRNAs in female oocytes in high density, to serve as a crucial basis for swarming and migration" (lines 370-372).

- 332- 361: These two sections in the Discussion repeat several of the same points. I suspect they could be edited down.

We thank the Reviewer for this suggestion. We now reduced the two sections and combined them into a single paragraph (lines 374-388).

- 338-339: The statement "Thus, response of parents to high-density signals is a crucial intrinsic regulatory mechanism for population homeostasis" suggests a sort of group selection adaptation that seems unwarranted. It seems more likely that this mechanism has evolved simply as females whose offspring experience less cannibalism contribute more to the future gene pool.

We thank the Reviewer for this comment. We revised the sentence according to this suggestion (lines 379-380).

- 350: Editing request: "In the current study, we report a pathway..."

We thank the Reviewer for this comment. We revised the sentence according to this suggestion (line 381).

- 353: Editing request: "...chemical stimuli to alter the behavior..."

We thank the Reviewer for this comment. We revised the sentence according to this suggestion (lines 383-384).

- 372-373: How many studies are being referenced here? There's only one citation at the end of the sentence.

We cited one reference here, to show that FOXN1 can serve as a maternal regulator of offspring. We now revised the sentence to "a previous study suggested that the *Drosophila* homologue of FOXN1 JUMU is a maternal regulator for zygote genome activation" (lines 405-407).

- 374: Editing request: "Therefore, our study reveals a previously unknown role of FOXN1..."

We thank the Reviewer for this comment. We revised the sentence according to this suggestion (lines 407-408).

- 389: Editing request: "In contrast, high density..."

We thank the Reviewer for this comment. We revised the sentence according to this suggestion (line 422).

- 392-393: "PTBP1-promoted nuclear export of pre-miR-276 is universally present in insects." The manuscript reports this interaction in two insect species. That cannot be called universal. (Especially if miR-276 is unique to insects!)

We thank the Reviewer for this comment. We revised the sentence to “PTBP1-promoted nuclear export of pre-miR-276 both in locusts and *Drosophila*) (line 440).

- Reporting Summary, Data section: I was unable to reach LocustBase at the URL provided.

We apologize for the technical errors of the database connection. We renewed the URL into <http://159.226.67.243> (line 560 and Reporting Summary).

References

1. He, J., et al. MicroRNA-276 promotes egg-hatching synchrony by up-regulating *brm* in locusts. *Proc. Natl Acad. Sci. USA* **113**, 584-589 (2016).
2. Zhao, L., et al. Phase-related differences in egg production of the migratory locust regulated by differential oosorption through microRNA-34 targeting *activin β* . *PLoS Genet.* **17**, e1009174 (2021).
3. He, J., et al. piRNA-guided intron removal from pre-mRNAs regulates density-dependent reproductive strategy. *Cell Rep.* **39**, 110593 (2022).
4. Nelson, C. E., Duvall, C. L., Prokop, A., Gersbach, C. A., Davidson, J. M. Chapter 29 - Gene delivery into cells and tissues. In: *Principles of Tissue Engineering (Fifth Edition)* (eds Lanza R, Langer R, Vacanti JP, Atala A). Academic Press (2020).
5. Hamilton, A. G., Swingle, K. L., Mitchell, M. J. Biotechnology: Overcoming biological barriers to nucleic acid delivery using lipid nanoparticles. *PLoS. Biol.* **21**, e3002105 (2023).
6. Zöllner, H., Hahn, S. A., Maghnouj, A. Lentiviral Overexpression of miRNAs. In: *miRNA Maturation: Methods and Protocols*. Humana Press (2014).

7. Hou, L., Li, B., Ding, D., Kang, L., Wang, X. CREB-B acts as a key mediator of NPF/NO pathway involved in phase-related locomotor plasticity in locusts. *PLoS Genet.* **15**, e1008176 (2019).
8. Kang, X., Yang, M., Cui, X., Wang, H., Kang, L. Spatially differential regulation of ATF2 phosphorylation contributes to warning coloration of gregarious locusts. *Sci. Adv.* **9**, eadi5168 (2023).
9. Kallio, P. J., Wilson, W. J., O'Brien, S., Makino, Y., Poellinger, L. Regulation of the Hypoxia-inducible Transcription Factor 1 α by the Ubiquitin-Proteasome Pathway. *J. Biol. Chem.* **274**, 6519-6525 (1999).
10. Matsuzaki, H., Daitoku, H., Hatta, M., Tanaka, K., Fukamizu, A. Insulin-induced phosphorylation of FKHR (Foxo1) targets to proteasomal degradation. *Proc. Natl Acad. Sci. USA* **100**, 11285-11290 (2003).
11. Yang, M., et al. MicroRNA-133 inhibits behavioral aggregation by controlling dopamine synthesis in locusts. *PLoS Genet.* **10**, e1004206 (2014).
12. Wang, J., et al. XPO5 promotes primary miRNA processing independently of RanGTP. *Nat. Commun.* **11**, 1845 (2020).
13. Bennasser, Y., et al. Competition for XPO5 binding between Dicer mRNA, pre-miRNA and viral RNA regulates human Dicer levels. *Nat. Struct. Mol. Biol.* **18**, 323-327 (2011).
14. Tamkun, J. W., et al. brahma: A regulator of Drosophila homeotic genes structurally related to the yeast transcriptional activator SNF2SWI2. *Cell* **68**, 561-572 (1992).
15. Shi, J., et al. Drosophila Brahma complex remodels nucleosome organizations in multiple aspects. *Nucleic Acids Res.* **42**, 9730-9739 (2014).

16. Dysart, R. J. Insect predators and parasites of grasshopper eggs. In: *Grasshoppers: Their Biology, Identification and Management User Handbook, United States Department of Agriculture, APHIS Technical Bulletin*. (1966).

17. Kutcherov, D. Egg-hatching synchrony and larval cannibalism in the dock leaf beetle *Gastrophysa viridula* (Coleoptera: Chrysomelidae). *Zoology (Jena)* **118**, 433-438 (2015).

REVIEWERS' COMMENTS

Reviewer #1 (Remarks to the Author):

My major concerns are addressed in the revised manuscript. I believe the manuscript is suitable for acceptance and publication in Nature Communications.

Reviewer #3 (Remarks to the Author):

Thank you for your thoughtful responses to the criticism raised by myself and the other reviewers. I am satisfied that the revised manuscript addresses these concerns.

I did note a small number of grammatical errors that can hopefully be corrected before publication.

- 67: "intensive" doesn't seem the correct word here. Perhaps "specific"?
- 78: "...in locust groups contributes to developmental..."
- 82: "While" rather than "As"
- 84: "remains elusive"
- 84: Remove "intricate"
- 85: "repsonds"
- 340: "The hatching synchrony within egg pods is a crucial factor..."

REVIEWER COMMENTS

Reviewer #1 (Remarks to the Author):

My major concerns are addressed in the revised manuscript. I believe the manuscript is suitable for acceptance and publication in Nature Communications.

We thank the Reviewer for this positive comments.

Reviewer #3 (Remarks to the Author):

Thank you for your thoughtful responses to the criticism raised by myself and the other reviewers. I am satisfied that the revised manuscript addresses these concerns.

I did note a small number of grammatical errors that can hopefully be corrected before publication.

- 67: "intensive" doesn't seem the correct word here. Perhaps "specific"?

We thank the Reviewer for this comment. We revised the sentence according to this suggestion (line 65)

- 78: "...in locust groups contributes to developmental..."

We thank the Reviewer for this comment. We revised the sentence according to this suggestion (line 77).

- 82: "While" rather than "As"

We thank the Reviewer for this comment. We revised the sentence according to this suggestion (line 80)

- 84: "remains elusive"

We thank the Reviewer for this comment. We revised the sentence according to this suggestion (lines 83).

- 84: Remove "intricate"

We thank the Reviewer for this comment. We removed the letters according to this suggestion (line 83).

- 85: "repsonds"

We thank the Reviewer for this comment. The subject of the clause is “miRNAs”, so we used respond here (lines 83).

- 340: "The hatching synchrony within egg pods is a crucial factor..."

We thank the Reviewer for this comment. We revised the sentence according to this suggestion (lines 339).